# ReVisE: Remote visualization environment for large numerical simulation datasets

**Stepan Orlov**[ID]@*, **Alexey Kuzin**[ID]@, **Alexey Zhuravlev**[ID]‡, **Vyacheslav Reshetnikov**[ID]‡, **Egor Usik**‡, **Vladislav Kiev**‡, **Andrey Pyatlin**‡

Institute of Machinery, Materials, and Transport, Peter the Great St. Petersburg Polytechnic University, St. Petersburg, Russian Federation

@ These authors contributed equally to this work.
‡ These authors also contributed equally to this work.
* majorsteve@mail.ru

**Data Availability Statement:** The source code of the ReVisE system is available in a public repository hosted on GitHub under the URL https://github.com/deadmorous/revise All datasets

## Abstract

The paper presents a new open-source visualization system, named ReVisE, aimed to provide interactive visualization of large datasets, which are results of complex numerical simulations. These datasets are hosted on a remote server or a supercomputer. The design of the system is briefly described. Dataset representation, proposed for interactive visualization and implemented in the system, is discussed. The effectiveness of our approach is confirmed by results of performance measurements on test and real-life large datasets. A comparison with other visualization systems is presented. Future plans of system development are outlined.

## 1 Introduction

Today's supercomputers are often used for numerical simulations of complex problems, such as aerodynamics, hydrodynamics, oil flow through porous media, and many others. Computational meshes (further referred to simply as meshes) used in those simulations can reach quite large size, e.g., $10^8$–$10^{10}$ nodes. Therefore, simulation results are represented by quite large datasets, especially in the case of unsteady problems, since the time adds one more dimension.

In this paper we address the problem of interactive visualization of large datasets resulting from complex simulations. Our contribution to the problem solution is software implementation of a new open-source visualization system, named ReVisE (Remote Visualization Environment), available at https://github.com/deadmorous/revise. The motivation behind the idea to create yet another visualization system is basically that existing widely used systems have serious performance issues, making truly interactive visualization only possible for relatively small datasets (up to $10^6$ nodes). Although systems for visualizing large datasets exist, e.g., NVIDIA IndeX (https://developer.nvidia.com/nvidia-index) and Sight [1], those are not widely used, in particular, due to their limited availability.

Our main goal is to design and a common way of data processing for visualizing fields on large meshes. This implies a practical proof that our approach to visualization is feasible. The current state of ReVisE software can be viewed as such a proof. It also shows that our

considered in the paper are available for download at https://ftp.mpksoft.ru/revise_datasets/ Step-by step instruction on how to use those datasets and repeat ReVisE performance measurements are available in two public protocols, which are as follows: Build ReVisE on Ubuntu https://www.protocols.io/view/build-revise-on-ubuntu-bruwm6xe dx.doi.org/10.17504/protocols.io.bruwm6xe Prepare and run test on available dataset https://www.protocols.io/view/prepare-and-run-test-on-available-dataset-bruzm6x6 dx.doi.org/10.17504/protocols.io.bruzm6x6.

**Funding:** This work has been funded by Russian Science Foundation URL: https://rscf.ru/en/ Grant No: 18-11-00245 Authors who received award: S. Orlov, A. Kuzin, A. Zhuravlev, V. Reshetnikov, E. Usik, A. Pyatlin The funders had no role in study design, data collection and analysis, decision to publish, or preparation of the manuscript.

**Competing interests:** The authors have declared that no competing interests exist.

visualization system can run (and give satisfactory results) on different hardware, from PCs to multi-GPU servers.

To give some context for our investigation, let us briefly discuss major existing software products for the visualization of large datasets.

Kitware ParaView [2] is an open-source multi-platform visualization system. It was developed to visualize extremely large datasets, with the ability to distribute data processing across many remote computational nodes. It can be deployed on both supercomputers and laptops (with lower capabilities). ParaView supports a mechanism to develop specialized plugins. An example of ParaView usage the system is presented in [3].

NVIDIA IndeX is another framework for remote visualization. It uses computing capabilities of GPUs to process big data. NVIDIA IndeX is designed for real-time visualization and can run on a GPU-accelerated cluster. In addition, a specialized plugin to ParaView has been developed. The plugin improves visualization performance for large datasets. It is claimed that the framework has the good scalability across GPU-accelerated nodes.

Sight [1] visualization tool is developed in Oak Ridge National Laboratory (ORNL) and is deployed on Oak Ridge Leadership Computing Facility (OLCF) systems (https://www.olcf.ornl.gov/olcf-resources/rd-project/sight/). It is used for needs of OLCF projects users and is intended to visualize large systems consisting of particles. It is built on client/server architecture, therefore the render server is on HPC cluster and only Web client is on the user's side. The render server supports rendering both on CPUs with OSPRay [4] and on GPUs with NVIDIA OptiX [5] backends.

One more tool for visualization of the scientific and engineering data is Tecplot 360 (https://www.tecplot.com/products/tecplot-360/), which is mostly used for the visualization of computational fluid dynamics (CFD) simulation results. This is commercial software developed by Tecplot company. The support for big data manipulation is provided by the SZL technology helping to reduce file sizes, processing times, and required operating memory. It is claimed that the visualization using of SZL is about 7.3 times faster and the peak memory usage is 93% less than with the traditional usage of the data files in the PLT format.

There are domain-specific visualization systems; one of them is Voxler (https://www.goldensoftware.com/products/voxler). It is used mostly by geophysicists, geologists, GIS professionals and other environment science researchers and engineers. Voxler provides specialized tools to visualize different 3D models, such as boreholes, LiDAR data, point clouds, etc. Its usage example is demonstrated in [6]. Another promising open-source visualization system is named FAST [7] and intended for heterogeneous medical image computing and visualization.

Among middleware related to the rendering of scientific datasets we would like to mention OSPRay [4] (an open-source library, based on a number of libraries by Intel, for parallel ray-tracing complex scenes on CPUs) and NVIDIA OptiX ray-tracing engine running on GPU [5].

A thorough review of big data visualization and analysis tools is presented in [8].

It is no surprise that attempts to directly process large amounts data resulting from a complex simulation fail to provide sufficient visualization performance of at least several frames per second (fps), even with the use of supercomputers. For example, the data containing several scalar fields for a single time layer may reach several gigabytes in size, so its fast processing would require hundreds or thousands of CPU cores, plus a very efficient data access solution, like Apache Hadoop/Spark. On the other hand, the image obtained as the output of visualization pipeline is just several megabytes in size. Therefore, we naturally face the question: do we really need so much (about 1 megabyte per pixel for a mesh with $10^8$ nodes) data from simulation results for visualization? In this paper, we are going to show that it is in fact not needed.

Further we describe the core technology in the basis of our approach to making interactivity possible even for visualizing large datasets, and the ReVisE system built on top of it.

The paper [9] presents a review of visualization techniques employing volume rendering algorithms, and suggests their categorization, aiming to analyze the problem of scalability with respect to the amount of input data and available computational resources for the visualization. An important idea presented in the paper is that the amount of data processing should depend on the visualization output; the terms *output-sensitive*, *ray-guided*, and *display-aware* are introduced to characterize visualization techniques that realize the idea to some extent. Further, the paper classifies data representation in terms of supporting concepts of bricking, octrees, multi-resolution hierarchies, and efficient storage layout and compression. Visualization strategies providing mechanisms to achieve high scalability are discussed. Those mechanisms include object-space and image-space decomposition, preprocessing, multi-resolution rendering, on-demand processing, and even ray-guided rendering, when the requests to load data come during volume ray casting. Those techniques help to fully utilize computational resources on the one hand, and provide a way to minimize the size of *active working set* on the other hand.

According to the categorization in [9], ReVisE is an output-sensitive visualization system, employing multi-resolution rendering based on octree data representation with additional bricking that results in a multi-resolution hierarchy of limited-depth octrees. The storage layout is optimized for consecutive reading in order to improve the performance. A number of formats are supported for source datasets, but those are always transformed into ReVisE format at the preprocessing stage.

The structure of next sections is as follows. Section 2.1 introduces the *sparse 3D mipmaping* technology lying in the core of ReVisE and responsible for data representation, including the data preprocessing stage; it also outlines octree visualization algorithms. Sections 2.2–2.4 present general architecture of the system, including a rendering server, a video streaming service, a web server, and a front-end running in the client browser. In section 3 the performance of the system is tested on a number of different-sized datasets and on different hardware. The results of performance tests and the quality of rendered images are further analyzed and discussed. Section 4.1 presents a comparison of interactive operation, animation, and rendering quality between ReVisE and ParaView. Section 4.2 outlines future work to be done, and section 5 summarizes results obtained.

## 2 Materials and methods

### 2.1 Sparse 3D mipmapping

The sparse 3D mipmapping technology has been proposed in [10] and implemented as part of the ReVisE system. This section partly repeats materials from [10] for the sake of easier understanding, additional details can be found therein.

The basic idea of Sparse 3D mipmapping is to remap all scalar fields defined on the original mesh onto a grid consisting of vertices of all cubes belonging to a sparse octree [11]. There are a number of benefits from this operation. In particular, sparse octree data can easily be transformed into a set of dense 3D textures and used for visualization employing volume rendering algorithms; those algorithms run quite fast on modern GPUs, partly because no geometry primitives need to be generated. In addition, octree data can easily be organized as a set of levels with fixed spatial resolution; due to that, progressive rendering is easily implemented, making visualization application quite responsive to user actions, while still delivering high quality images in additional time. Finally, the size of the octree dataset can be controlled by limiting the maximum spatial resolution.

**2.1.1 Using octree for scalar field representation.** Octree data structure can be used to represent a spatial scalar field in the following way. Suppose there is an original mesh—unstructured or structured, consisting of tetrahedra or hexahedra—it does not matter, and a scalar field specified at mesh nodes. Mesh elements provide a way to interpolate field values at arbitrary points of the domain. Importantly, unstructured meshes are often non-homogeneous, in the sense that mesh element size varies significantly across the domain. For a given mesh, an octree can be generated, such that the leaves of the octree have sizes close (in some sense, which is clarified in Subsection 2.1.2) to the sizes of elements they intersect with. We do not use the octree to store any data in its nodes. Instead, we use *grids induced by the octree* (or its part) to represent scalar fields. A given octree induces the grid whose nodes are vertices of all octree cubes, and whose elements are octree cubes. Notice that the induced grid is often sparse because the original mesh is non-homogeneous and because the domain differs from the cube constituting the octree root.

A scalar field can be interpolated in each node of the induced grid that belongs to the domain, resulting in a real number. Nodes of the induced grid that lie outside the domain are marked by storing NaN (not-a-number) value. As a result, the field is represented as an array of real numbers (some having NaN value). To associate each element of the array with a particular node of the induced grid, an ordering rule has to be established for induced grid nodes. A simple ordering rule can be obtained by recursive depth-first traversal of the octree, assigning successive numbers to the unnumbered vertices of the current cube; other ordering rules could be considered, e.g., those that can be run in parallel. Further we refer to fields defined at nodes of the induced grid as *sparse fields*.

Once a sparse field is computed, it can be interpolated at a subset of domain points: the interpolation is possible when the smallest cube containing the point does not contain induced mesh nodes with NaN field value. Our implementation of the interpolation algorithm computes field values on the regular *dense grid* (obtained from sparse grid by inserting nodes) and interprets the resulting array of values as a 3D texture at the visualization stage.

Fig 1 illustrates the representation of a field using quadtree in the 2D case, which is similar to the 3D case.

It is worth noticing that when the elements of the original mesh are large enough, the octree nodes are equally large too, which may lead to noticeable "marginal area" near the domain boundary, where field cannot be interpolated (Fig 1D). To overcome this problem, our octree generation algorithm has a part employing the "virtual refinement" of domain boundary in order to reduce the marginal area. A parameter controls the relative density of refined boundary mesh. As a result, the leaves of the octree become small enough near the domain boundary, which ensures sufficient visualization quality. Fig 1E shows an example of quad tree refined near the boundary, and Fig 1F—the resulting scalar field.

**2.1.2 Controlling the size and quality of visualization dataset.** As mentioned above, the leaves of generated octree are "close" in size to elements located nearby. It is important to define that closeness in a proper way, because different definitions may lead to very different octree sizes. Importantly, meshes used, e.g., in computational fluid dynamics are often anisotropic, which means they contain very thin elements needed to model boundary layers: orthogonal edges of a hexahedral mesh element may differ in length 100 times and more, as shown in Fig 2B.

The algorithm of octree generation iterates through mesh elements; for each element, it computes the axis-aligned bounding box and estimates element size. Then it picks closest possible size for octree node and, if necessary, inserts octree nodes to ensure that each point of the element bounding box belongs to an octree node of the chosen size. The element size $s$ is bounded between two values, $s_{\min}$ and $s_{\max}$. On the one hand, the smallest element edge length,

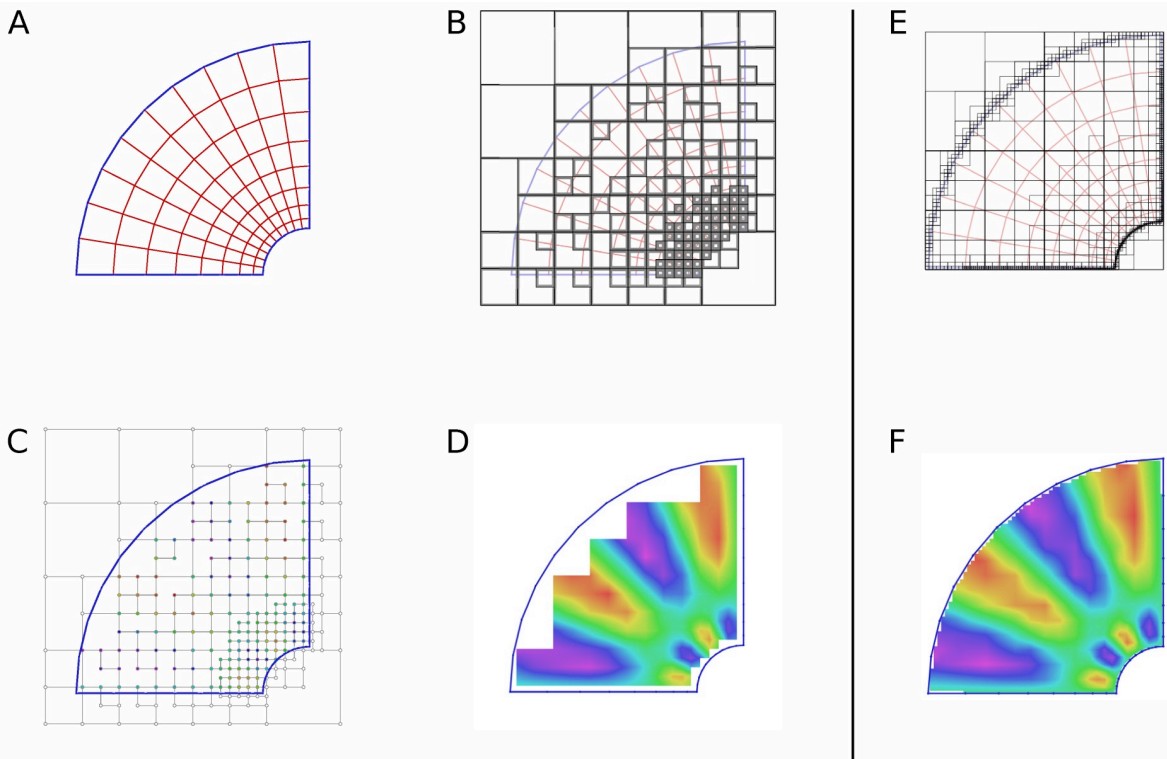

**Fig 1. Using quad tree for scalar field representation.** A quadtree is generated according to element size distribution in the original mesh. Scalar fields are interpolated at nodes of the sparse grid induced by the octree, then field can be interpolated at subdomain points excluding the marginal area. The size of the marginal area can be reduced by refining the octree near the boundary. A: Original mesh. B: Generated quadtree. C: Field interpolated at nodes of the induced grid. D: Field interpolated at subdomain points. E: Quadtree refined at boundary. F: Field interpolated at subdomain points in the case of refined boundary.

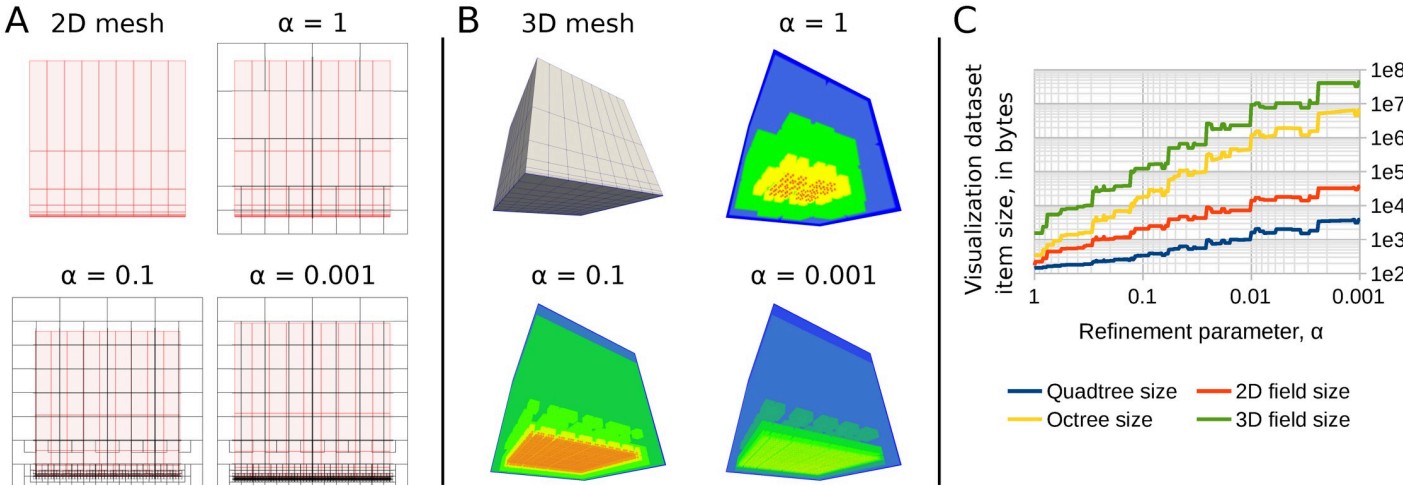

**Fig 2. Quadtree/Octree generation on anisotropic meshes.** Anisotropic meshes contain thin elements having very different edge sizes. Element size estimation is important for controlling the size of visualization dataset. A: 2D mesh and quadtrees generated for it with different $\alpha$. B: 3D mesh and octrees generated for it with different $\alpha$ (smaller octree nodes "shine through" bigger ones). C: Dependency of the octree size and the sparse field size on $\alpha$.

$s_{min}$, should not be considered element size: for thin elements at the boundary, this would lead to a huge number of octree nodes, because other element edges are much longer. On the other hand, taking the largest element edge length, $s_{max}$, as element size, might lead to a too coarse octree. Our approach is to take the value $s = s_{min} + \alpha(s_{max} - s_{min})$ as the element size, where $\alpha \in [0, 1]$ is a constant parameter. Positive values of $\alpha$ help to prevent the "explosion" of octree size due to thin mesh elements.

Fig 2 shows the results of quadtree and octree generator test runs on simple 2D and 3D meshes respectively, containing thin elements at one face. Practically, the parameter $\alpha$ is chosen between 0.1 and 1 to trade-off visualization quality against the size of sparse field, which is typically comparable or less than the size of the field on the original mesh.

Another idea is to limit the total depth of the octree to a fixed level $d_0$, thus limiting maximum possible octree size. Practically reasonable values of $d_0$ are between 10 and 13 and correspond to the maximum resolution between 1024 and 8192 smallest octree nodes in each dimension, which is enough in many practical cases.

**2.1.3 Visualization metadata.** As already mentioned, at the visualization stage sparse fields are turned into dense fields represented by 3D textures, which are further rendered on GPUs. But the memory of a single GPU is limited and can only store 3D texture corresponding to an octree of depth no more than 9 or 10, depending on hardware. In the same time, the original mesh can have local refinement that results in sparse octrees of bigger depth. Besides, it is promising to parallelize the rendering across many GPUs. Therefore, a sparse field has to be separated into several smaller sparse fields that can be processed in parallel.

To achieve this goal, we introduce *metadata* of the visualization dataset. The metadata is constituted by the octree, the block depth parameter $d$, and the *level structure*. Let us define the term *block* as a subset of the octree nodes, consisting of a node (called block root) and all its children up to depth $d$ from block root. Thus, a block itself is an octree of depth at most $d$. Metadata level structure consists of blocks. Level $l$ consists of blocks whose roots are at distance $l$ from the octree root. Therefore, level 0 contains one block whose root is the octree root. Next levels have more blocks. A new level provides a more detailed representation of a scalar field if it has at least one block of depth $d$. Therefore, levels 0, 1, . . ., $D - d$ are necessary to represent an octree of depth $D$. Fig 3 shows an example of metadata for the 2D case.

In our current implementation, each level has to contain all $2^{3l}$ blocks, and the number of levels is limited to some $L$ (e.g., $d = 8$, $L = 3$, which allows to represent an octree of depth at most 11); to make it possible, we force the octree to have all children up to depth $L$, which has a relatively small storage overhead.

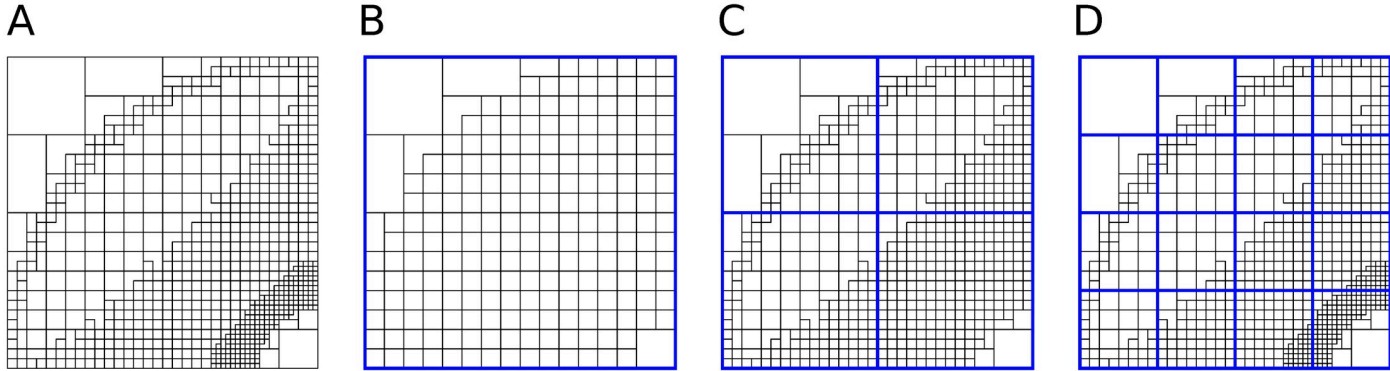

**Fig 3. An example of quadtree and corresponding metadata.** The quadtree has depth 6, and the metadata contains three levels; blocks of each level have depth at most 4. Level blocks are marked with blue squares. A: Quadtree. B: Metadata level 0 (single block). C: Metadata level 1 (2×2 blocks). D: Metadata level 2 (4×4 blocks).

Sparse fields are computed per-block, and for each scalar field a single file stores sparse fields for all blocks. In addition, metadata stores the location of sparse field for each block in the file.

This metadata structure allows to easily organize the rendering of a specific level and have the corresponding level of detail in the resulting image. Importantly, sparse field data can be read from a storage system in one operation for the entire level, eliminating the need for seek operations or for reading unnecessary data.

Practically, an application may choose between rendering a fixed level or doing the progressive rendering to provide a low-detailed image as fast as possible and provide more detailed images as they become available. Importantly, each block can be rendered independently on an SMP machine with multiple GPUs, or on in a distributed environment, such as a cluster.

**2.1.4 Octree storage optimization.** Although an octree is a popular data structure, and a number of methods [12–16] have been proposed to optimize its storage, we provide our own solution. Existing methods seem to be quite redundant or inappropriate for our case due to two reasons. Firstly, most often, an octree node is considered to either have all 8 child nodes or be a leaf, i.e., have no children at all. In the ReVisE system, we use a different flavor of this data structure, allowing each node to have any subset of child nodes. Secondly, we do not need to store data at octree nodes, as others do. Instead, as explained above, data items are stored at nodes of induced grids that need to be extracted from the octree.

According to previous sections, there are two basic usage patterns of the octree data structure. At the octree generation stage, its nodes need to be located by position in space, and new nodes need to be added. At the field interpolation and visualization stages, nodes need to be located by position, and induced grids need to be extracted from limited-depth subtrees. Importantly, the octree is immutable after it is generated; this gives a way to drastically reduce disk and operative memory usage for storing and operating octree.

At the octree generation stage, it is represented by root node geometry parameters (edge length $L$ and center position $\mathbf{R}_c$) and a single array containing octets of integer indices. Each octree node has an ordinal number, and the ordinal of the root node is 0. Each octet $C_i$ corresponds to $i$-th octree node and contains ordinal numbers of child nodes or zeros indicating the absence of a child.

For large-scale meshes, the size of the corresponding octree in memory might be quite large. It usually fits into memory for a single subdomain, but the processing of the entire mesh might lead to unacceptably large size.

Fortunately, the octree data structure can be stored much more efficiently if there is no need to add new nodes. The basic idea is to change ordinal numbers of octree nodes in such a way that they appear in the octet in the ascending order. Therefore, the required storage capacity reduces from 32 or 64 bits necessary to store an integer index to just one bit. During the consecutive reading of the octree, actual indices of child nodes can easily be restored. In addition, all trailing zeros can be omitted, which further reduces the storage size up to 8 times (that is the limit case for full octree). The octree stored in this way is further referred to as *the compressed octree*. Importantly, the compressed octree with search and recursive walk enabled typically fits into 1–20 Gb of random access memory (RAM) even for meshes with $10^{10}$ nodes.

This algorithm of octrees compression is described in more details in [10].

**2.1.5 Preprocessing simulation results.** Within our approach, the data has to be preprocessed before visualization. This needs to be done only once per simulation, and the preprocessing is fully automatic. We did not focus on preprocessing performance, although its duration is reasonable. At the preprocessing stage the original dataset is transformed into the *visualization dataset*.

The original dataset is assumed to be distributed across nodes of a computer system, further referred to as *data host*. Each node of the data host stores data for a specific subdomain, constituted by corresponding part of the mesh (there are a number of domain decomposition techniques, e.g. those implemented in [17]). The preprocessing takes place on the nodes of the data host; subdomains are processed separately, and further the subdomain data is merged into a single visualization dataset. The preprocessing consists of the following five stages: (1) the calculation of the bounding box containing the entire mesh; (2) octree generation at each subdomain; (3) the interpolation of scalar fields at each subdomain; (4) merging all octrees into one; and (5) merging scalar fields in such a way that they are defined at sparse grids induced by the global octree.

In order for subdomain parts of visualization dataset to be mergeable into a single dataset, the octree for each subdomain has to be representable as a subset of the global octree. Therefore, the preprocessing starts with computing the bounding box for the entire domain in order to determine the position and size of the cube at the root of the global octree. At stages 2 and 3, this information is necessary for the generation of "compatible" subdomain octrees. Stage 4 merges subdomain octrees pairwise. An algorithm for stage (4) has been proposed in [10] that is applicable to octrees in the compressed format (see previous section) without having to perform node insertion operations, because that would require much more memory. Modest memory requirement gives the proposed algorithm an advantage over other algorithms, such as [18]. For stage (5), a straightforward algorithm can be implemented that merges subdomain sparse fields pairwise. At the moment of paper writing, stages 4 and 5 are not implemented in software and are parts of work in progress.

**2.1.6 Visualizing octree data.** The visualization in the ReVisE system is based on volume rendering algorithms that require 3D textures to read field data. Before a rendering algorithm can be run, a sparse field has to be transformed into the corresponding dense field and stored in a 3D texture in GPU memory. Our implementation of this operation employs CUDA to generate the 3D texture with dense field on the GPU. For interactive visualization, the time spent on each operation is important. Since the rendering is done on GPU, one has to have 3D texture in GPU memory. On the other hand, data transfer between a GPU and the host system costs time and has to be minimized. Taking this into account, it sounds reasonable to upload a sparse field to the GPU and then generate the dense field on the GPU. Also it should be noticed that sparse fields are often 10–100 times less in size than corresponding dense fields.

The algorithm of dense field generation processes the octree block of depth $d$ level by level, starting from root and ending at level $d - 1$. To process each level, a CUDA kernel is invoked once. When level $l$ is processed, the algorithm assumes that the dense field is correct on the grid induced by the full octree of depth $l$ (this is always true for level 0). The objective of kernel running on GPU is to ensure the dense field is correct on the grid induced by the full octree of depth $l + 1$. To imagine the algorithm doing that, consider leaves of a full depth-$l$ octree. The geometry of each leaf is a cube. The algorithm interpolates any missing field values in the middles of all cube edges, then in the middles of all cube faces, then in the middles of all cubes. Since the dense field is known at level $l$, it is known at all cube vertices, and the above mentioned interpolation is done easily. An important thing to keep in mind is a special handling of sparse field NaN values indicating that the corresponding points are outside the domain. When such a value is involved in the interpolation, the result of the interpolation has to be NaN. This simple propagation rule for NaN values ensures the correctness of dense field in the domain, excluding the marginal area (see Fig 1D and 1F).

Once the dense field is computed, it is interpreted as a 3D texture and used by rendering algorithms of volume raycast type [19], implemented in CUDA. Our implementation of the rendering algorithms is based on source code of OpenGL fragment shaders found in the

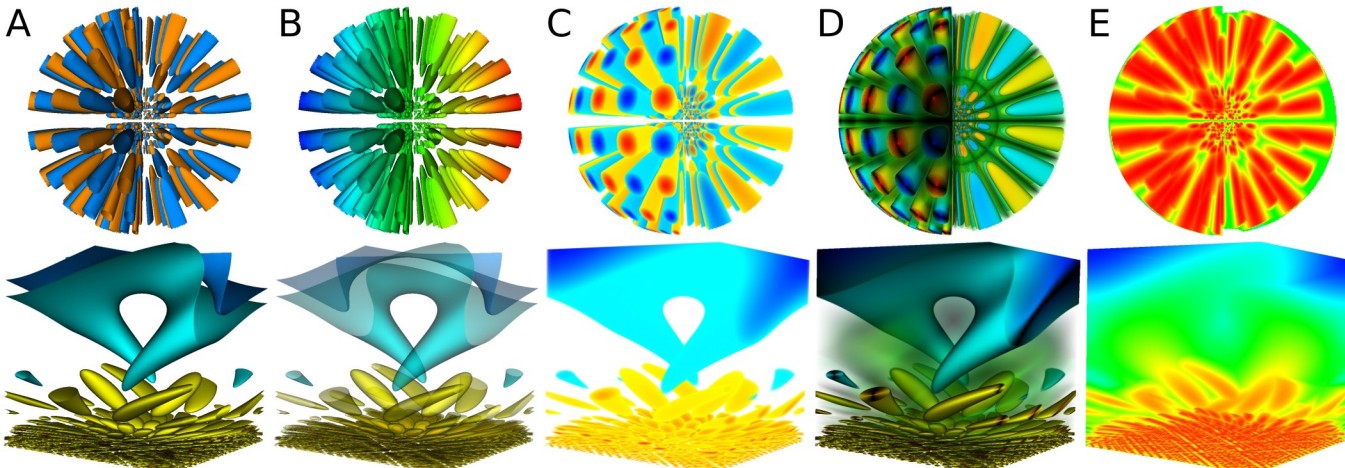

**Fig 4. Volume rendering algorithms applied to test datasets.** A: Opaque isosurfaces. B: Field values on isosurfaces of other field; transparent isosurfaces. C: Colormap with transparency. D: Colormap with transparency and per-sample lighting. E: Maximum intensity projection.

Visualization Library [20], and extends the functionality of the original shaders. Some examples of algorithms currently implemented are shown in Fig 4. A rendered image can be attributed to certain metadata level; it is obtained by rendering all level blocks. On multiple GPU systems—including clusters equipped with GPUs—the set of level blocks is split into subsets, each of which is rendered by one GPU. Image parts rendered by each GPU are then blended back-to-front into a final image. To make this approach work, the separation of level block set into subsets has to ensure that subsets can be ordered for correct back-to-front blending. Besides, similar blending takes place within each subset; for this to work, blocks of each subset have to be ordered for back-to-front blending. For a fixed level number and a particular number of GPUs and CPUs available, the rendering algorithm can be represented as a graph of tasks; each task can be executed as soon as all its inputs are available and the corresponding computational resource (CPU or GPU) is waiting for a new task. An example of rendering task graph is shown in Fig 5.

## 2.2 General architecture of visualization system

Sparse 3D mipmapping technology lies in the basis of the ReVisE visualization system. The system is a distributed application that performs visualization on a remote system and provides web interface for clients to control and observe the visualization. Visualization system runs code of web server written in node.js; visualization code is written in C++ and is controlled from a native node.js add-on based on the node-addon-api technology. Visualization code is parallelized for use on SMP machines with single or multiple NVIDIA's GPUs; further plans include developing a parallel version for clusters. Rendering functionality, including parallel execution of the rendering task graph, is available as a separate library.

Apart from the web server that handles the interaction with client over HTTP, the system contains a video streaming service. The service runs in a separate thread of the web server process and is responsible for serving the sequence of images rendered; the images are transported using the web sockets protocol. Similarly to visualization code, the video streaming service is controlled by web server through an interface exposed basing on node-addon-api.

Client code runs in a web browser on the client side. It provides user interface elements for choosing among available problems, setting up visualization parameters, moving camera,

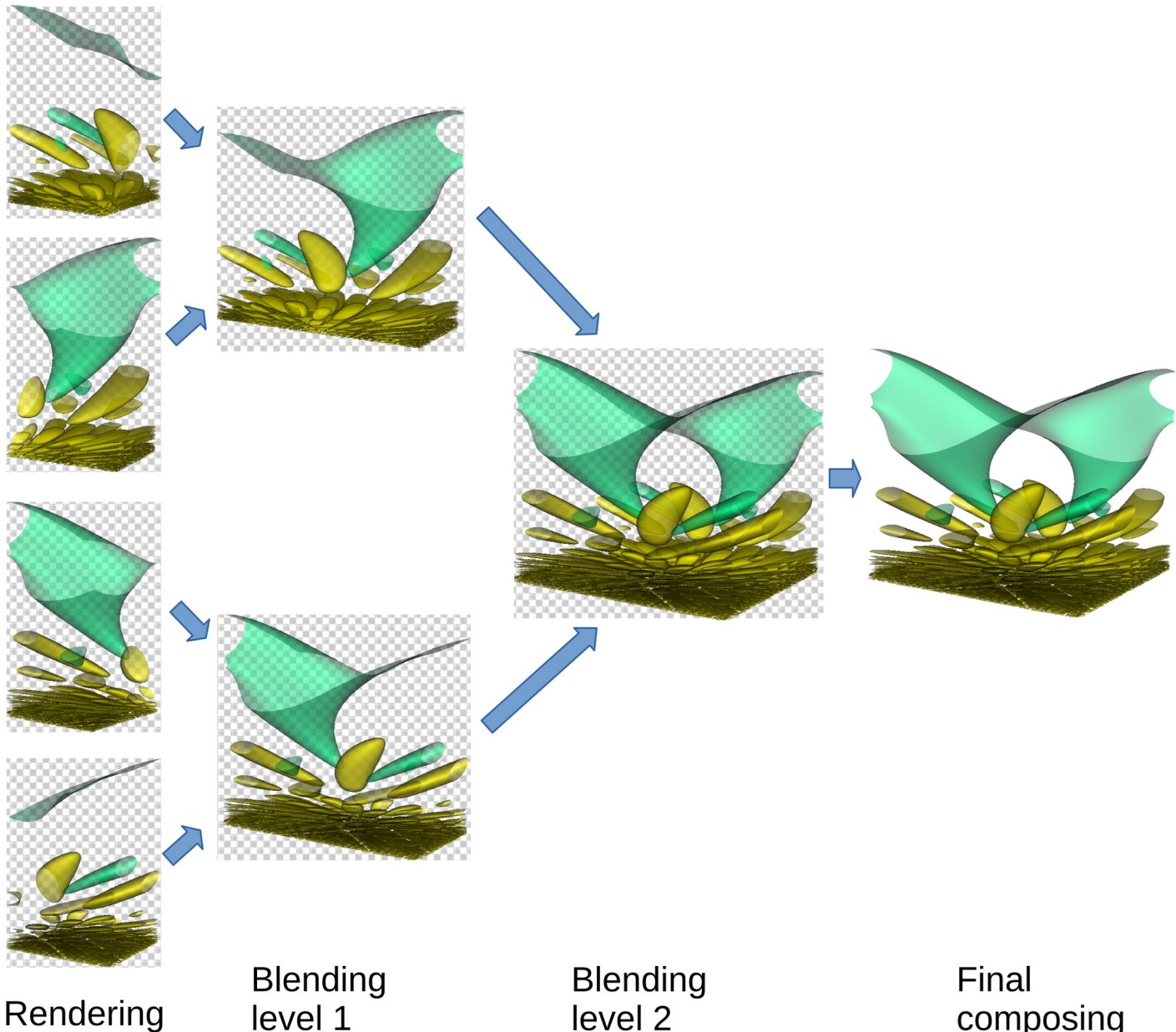

Rendering   Blending level 1   Blending level 2   Final composing

**Fig 5. An example of rendering task graph.** The graph consists of 4 parallel rendering tasks, two levels of image parts blending and final composing, where the background color is added.

navigating the timeline (including animation), and sharing video stream. No rendering takes place on the client side, as the client obtains images rendered by the server.

The architecture of the system is depicted in Fig 6.

## 2.3 Video streaming service

Once the visualizer renders a new final image, it uploads the image to shared memory; the image is accompanied by a header containing frame ordinal number and some additional

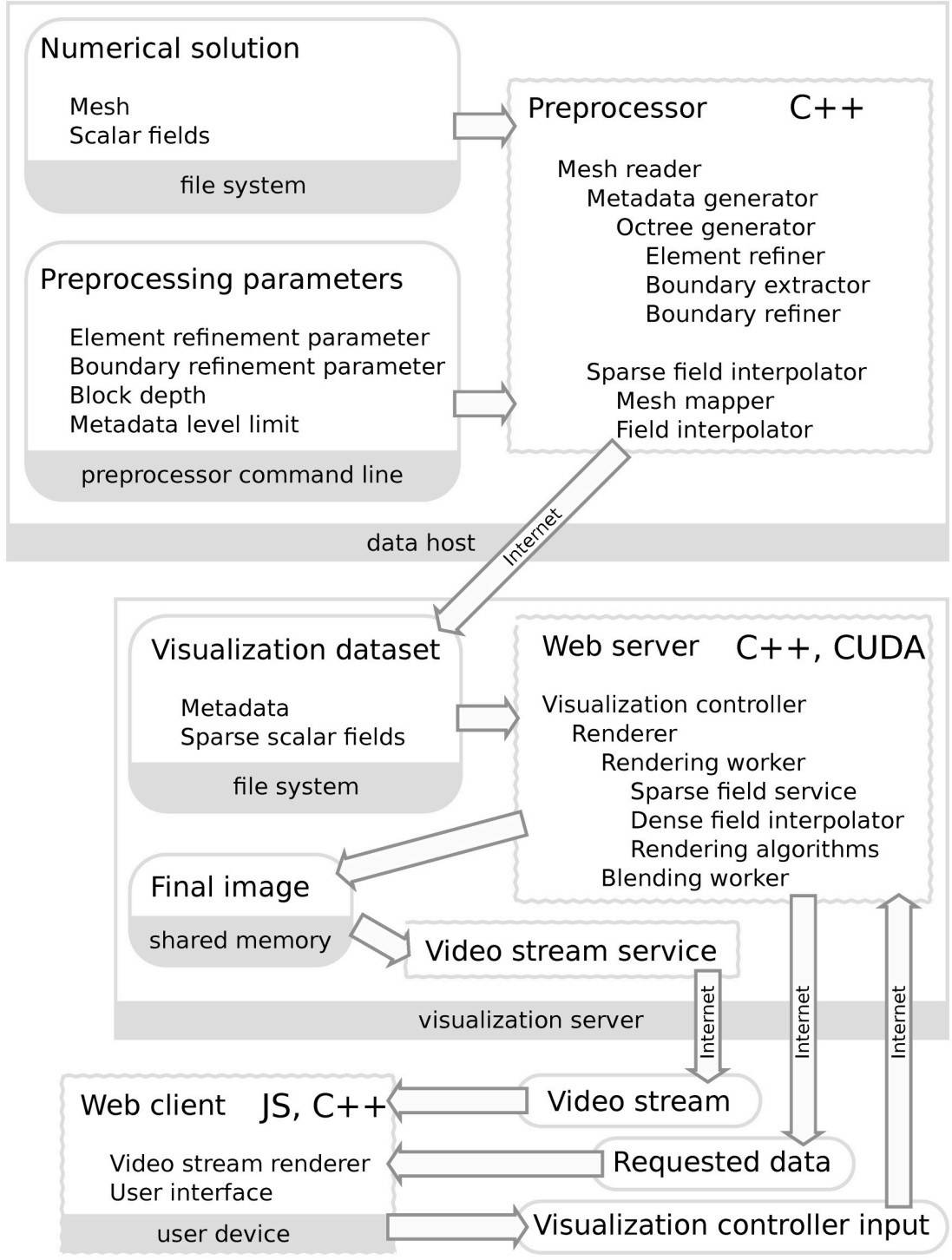

**Fig 6. General architecture of the ReVisE system.** Automatic preprocessing takes place on the data host; the resulting visualization dataset is copied to the visualization server. Interactive visualization application serves video stream of rendered images and listens to user input.

information, such as rendering duration and metadata level number. Image data is stored in shared memory without any compression.

Video streaming service checks shared memory contents periodically. When it finds a new frame number in the header, the frame is sent to subscribed clients over connected web sockets. Importantly, the streaming service needs to deliver new frames to the client as fast as possible, otherwise the visualization cannot be interactive. There are a number of approaches to implement interactive video streaming; some of them are discussed in [21] (unfortunately, details of communication protocols used over web sockets are not described). In [22], scaling video streaming across a number of clients is considered. In general, it appears to be tricky to deliver a low-latency video stream to the user in the case of a network with limited bandwidth and a noticeable latency.

Because visualizers are created by the web server per-client, there can be several shared memory areas for use as video sources. Once a new visualizer is created for a new client, or a visualizer is removed when a client is gone, the web server informs the streaming service about the change of set of available sources. When a client connects to video streaming service, it passes source identifier (previously obtained from web server), in order for service to know which source to use for the client.

Once connected, the client and the service follow a simple protocol, outlined below. Important features of the protocol are the abilities to trade-off image compression quality against image size and to overcome network latency to some extent.

The protocol specifies a number of messages; some of them are initiated by the client, and some—by the service; each message expects a reply message. Table 1 lists and briefly explains the messages.

Once the client specifies the source by sending the s message, the video stream service starts sending new images using the n message. A new image is sent when all of the following conditions hold: (1) time passed since last image was sent is not less than 40 ms; (2) the number of pending images (those sent in n messages for which replies have not been received yet) does not exceed some constant value (typically 3–10); (3) ordinal image number in source shared memory, $N_v$, is greater than the corresponding number in the header of most recent image sent to the client.

In addition, the streaming service keeps track of "overflow" events: when the conditions (1) and (2) both hold, it records overflow event status 0. When the condition (1) holds and (2) does not, it records overflow event status 1. A buffer of most recent 20 overflow status values is used to adjust the quality $Q \in [0, 100]$ of JPEG images sent to the client: once the average status value exceeds a constant threshold (currently 0.5), current image quality is decreased by some step (currently 1), unless it has reached minimum value $Q_0$ (currently 10); once the average

**Table 1. Messages specified in the video streaming service protocol.**

| initiator | message | reply message | remarks |
|---|---|---|---|
| client | s: ⟨sourceId⟩ | Ok / Unknown source | Instructs service to select the specified source |
| client | F | F: ⟨hdr⟩⟨img⟩ | Request full quality image |
| client | B | Ok | Prefer binary image data |
| client | T | Ok | Prefer base64-encoded image data |
| service | n: ⟨hdr⟩⟨img⟩ | n: ⟨$N_v$⟩⟨$N_s$⟩ | Service sends new image to client initiatively |

*sourceId* is a string identifying video stream source. *hdr* is the image header consisting of five numbers: $N_v$ (ordinal number of image rendered by visualizer), $N_s$ (ordinal number of image sent by the service to the client), *l* (metadata level corresponding to the image rendered), *T* (rendering task graph execution time), and *Q* (image quality). *img* is image data, in JPEG format (base64-encoded in case of text data).

status value is below another constant threshold (currently 0.4), current image quality is increased by the same step, unless it has reached maximum value 100. This simple algorithm allows to keep reasonable frame rates, on the one hand, and keep reasonably small delays between user actions and visualized images corresponding to those actions, even for low-bandwidth network connections and networks with latency 100–300 ms.

Client code receives and renders images sent with the n message in a straightforward manner; it should send reply message immediately after receiving an image, before rendering it. In addition, sometimes client requests a full-quality image using the F message. This happens, for example, when the last received image has quality lower than 100, and the time passed since receiving that image exceeds some threshold, e.g. 1 second.

## 2.4 Web application for remote visualization

The web application is a web server that runs on the visualization server and provides functionality to remotely control the visualization of datasets from a web browser. The server runs a separate copy of visualizer for each client session and provides web API to operate it. HTML pages sent to the user contain client code that interacts with the visualizer in response to user actions, such as the choice of an item from a list, setting a numeric parameter, mouse clicks, and mouse movements. This allows to track all input data for visualization controller, including camera position. As a result, user can fully control the visualization from the web browser. Fig 7 shows a screenshot of HTML page with user interface and a visualized scene.

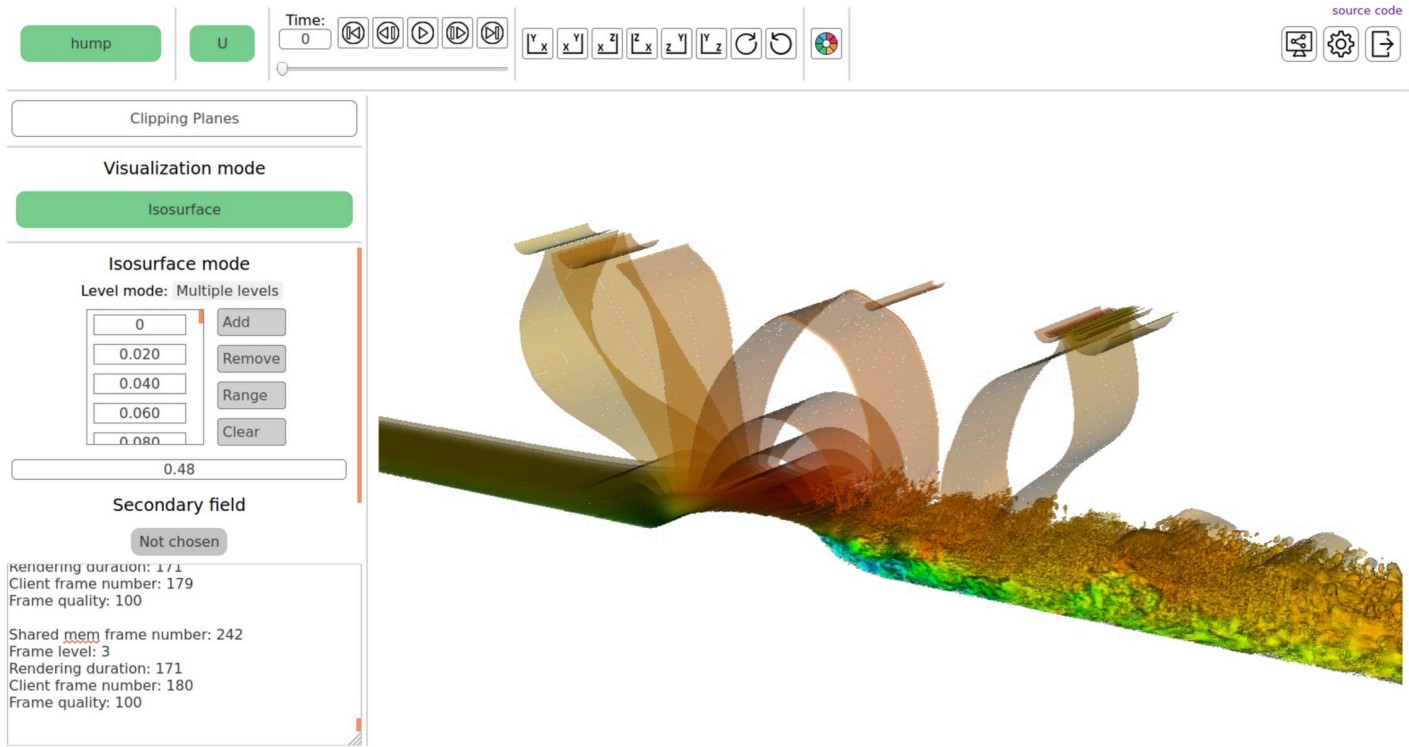

**Fig 7. ReVisE web interface.** Area at the top contains drop-down lists to choose a problem and a primary field, timeline navigation elements, camera positioning buttons, video stream sharing and settings buttons. Left panel contains elements to pick visualization algorithm and control its parameters. Central area displays rendered scene. Mouse can be used to pan, zoom, and rotate the scene interactively.

## 3 Results and evaluation

The ReVisE system has been tested in a number of problems of different scale, including test datasets and real-life problems. First of all, we focused on the visualization performance, aiming to provide as much interactivity as possible; preprocessing times have also been measured. Problem cases presented below are as follows.

1. Multiscale waves, further referred to as `mwaves`—a fully synthetic dataset providing an animated field computed by the formula

$$f(x, y, z, t) = \sin\frac{\pi x'}{2} \sin\frac{\pi y'}{2} \sin\left(\frac{1}{z} + \pi t\right),$$

   where $x' \equiv \cos\frac{\pi z}{2} x + \sin\frac{\pi z}{2} y$, and $y' \equiv -\sin\frac{\pi z}{2} x + \cos\frac{\pi z}{2} y$; $x, y, z$ are Cartesian coordinates, and $t$ is the time. Spatial domain is the cube $x \in [-1, 1]$, $y \in [-1, 1]$, $z \in [0.05, 2.05]$. The field has features with size changing gradually between two opposite faces of the domain (bottom row in Fig 4). The dataset is computed on the regular grid, which results in a full octree—the worst case for visualization performance.

2. Simulation of flow around 2d NASA wall-mounted hump [23], further referred to as `hump`. The mesh contains $5.26 \cdot 10^6$ nodes; numerical solution contains 100 time steps.

3. Simulation of flow over high-lifted turbine cascade at low Reynolds numbers [24] using NOISEtte code [25], further referred to as `cascade`. The mesh contains $8.98 \cdot 10^7$ nodes; subset of numerical solution for visualization contains 13 time steps.

For those problems, visualization datasets have been obtained and visualization performance has been measured. For performance tests, we used three systems, as shown in Table 2. Machine names `DGX-1`, `Tesla` and `GeForce` are used for reference and should be treated as identifiers. `DGX-1` and `Tesla` are multi-processor SMP machines and `GeForce` is a usual desktop.

`DGX-1` and `Tesla` were used in the remote server mode: they have only visualization server running, as shown in Fig 6, and interact with remote client via Internet. In the case of `GeForce`, the visualization server and web client were running locally.

The procedure of performance measurements is described in the following protocols: "Build ReVisE on Ubuntu" (https://dx.doi.org/10.17504/protocols.io.bruwm6xe), "Run ReVisE from Docker" (https://dx.doi.org/10.17504/protocols.io.bv5hn836), and "Prepare and run test on available dataset" (https://dx.doi.org/10.17504/protocols.io.bruzm6x6).

**Table 2. Parameters of the computers used for tests of ReVisE.**

|  | DGX-1 | Tesla | GeForce |
|---|---|---|---|
| CPU info | Intel(R) Xeon(R) CPU E5-2698 v4 @ 2.20GHz | Intel(R) Xeon(R) CPU E5-2650 v4 @ 2.20GHz | Intel i7-8700 CPU @ 3.20GHz |
| CPU count | 2 | 2 | 1 |
| cores per CPU | 20 | 12 | 6 |
| RAM, Gb | 512 | 128 | 16 |
| GPU info | V100-SXM2 GPUs (16 GB memory, 5120 CUDA cores per GPU) | Tesla V100-PCIE GPUs (32 GB memory, 5120 CUDA cores per GPU) | GeForce 1060 GPU (6 GB memory, 1280 CUDA cores) |
| GPU count | 8 | 2 | 1 |
| remarks |  | Intel Optane 960 storage system |  |

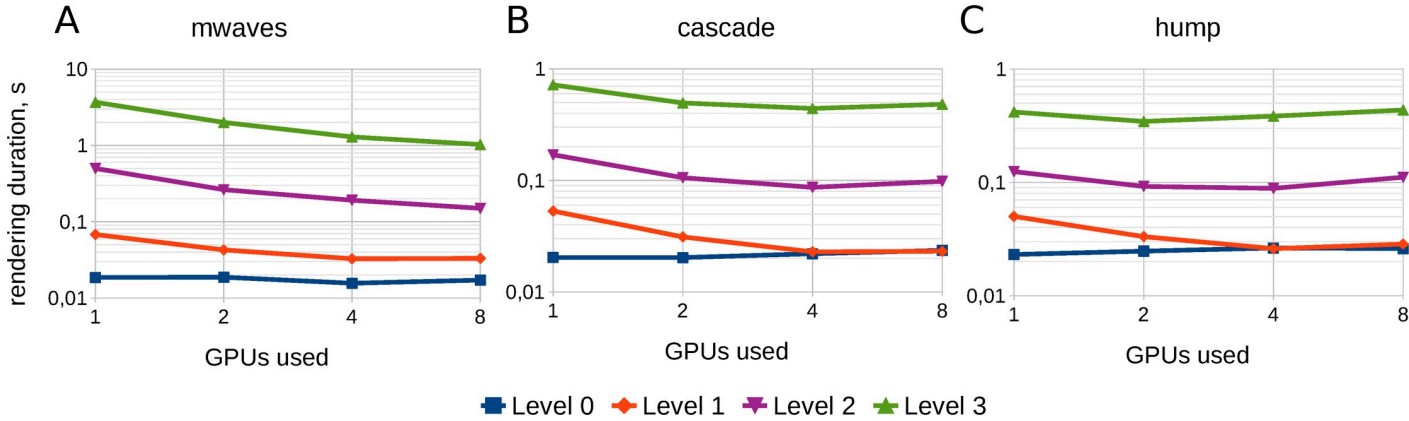

**Fig 8. Rendering duration on `DGX-1`.** Rendering duration as a function of GPU count used for problem cases described above. Rendering algorithm is maximal intensity projection. Each worker thread uses its dedicated GPU.

## 3.1 Visualization performance

Here visualization performance tests are described. These tests involve the assessment of rendering speed and scalability. The results of testing are presented in Fig 8. The chart contains the dependency of rendering duration on the number of GPUs used. The rendering is performed on `DGX-1`, for the maximal intensity projection rendering algorithm. The rendering is performed in parallel, and each worker thread is placed on a separate GPU. Therefore, the number of GPUs is equal to the number of worker threads and the case of one GPU corresponds to the sequential rendering.

The rendering of each dataset was performed separately for levels 0–3. Dataset parameters are presented in Table 3. The blocks of datasets `hump` and `cascade` have depth 8, while the blocks of `mwaves` have depth 7. But the dataset `mwave` is dense, therefore, this is the heaviest case with the biggest number of nodes. For instance, the total number of nodes at level 3 is $7.72 \cdot 10^6$ for `hump`, $7.96 \cdot 10^7$ for `cascade` and $1.1 \cdot 10^9$ for `mwaves`.

The number of blocks at each level is a power of 2 (1, 8, 64 and 512 blocks for levels 0–3 respectively), therefore, parallel rendering at 1, 2, 4 and 8 GPUs is ideally balanced, e.g. each worker thread processes the same number of blocks (of course, the density of the blocks may be different and it can produce an imbalance).

Also it is worth noticing that the dataset at level 0 consists of a single block, therefore the rendering at level 0 is always sequential. Theoretically, it must be a horizontal straight line; as one can see, that is actually the case.

The dataset at level 1 consists of 8 blocks, therefore, in the case of 8 GPUs used, each worker thread processes a single block. So one can expect that the rendering speed in this case is almost equal to the rendering speed at level 0. As one can see, this actually takes place in the

**Table 3. Parameters of visualization datasets.**

| | blocks depth, $d$ | total node count at level | | | |
|---|---|---|---|---|---|
| | | **0** | **1** | **2** | **3** |
| `mwaves` | 7 | $2.15 \cdot 10^6$ | $1.72 \cdot 10^7$ | $1.37 \cdot 10^8$ | $1.10 \cdot 10^9$ |
| `cascade` | 8 | $3.34 \cdot 10^5$ | $2.54 \cdot 10^6$ | $1.75 \cdot 10^7$ | $7.96 \cdot 10^7$ |
| `hump` | 8 | $2.02 \cdot 10^5$ | $1.46 \cdot 10^6$ | $6.39 \cdot 10^6$ | $7.72 \cdot 10^6$ |

case of `cascade` and `hump` problems, which means that the scalability of rendering level 1 is good and close to ideal. The result for `mwaves` is slightly worse, but also acceptable.

The results for levels 2 and 3 have scalability issues, especially in the cases of `cascade` and `hump`. The rendering time stays almost the same as the number of GPUs used grows. A possible cause of this issue could be the following.

As mentioned above, the whole rendering task consists of parallel rendering of image parts on GPU and image blending on CPU. Image blending appears to run rather fast, occupying about 5% of overall rendering time. Therefore, image blending time can be neglected, and attention should be paid to the time spent by rendering workers on GPU.

The performance of rendering tasks was investigated with the NVIDIA `nvprof` tool on the `Tesla` machine for the `hump` problem. Tests have been done for the case of rendering on a single GPU (sequential rendering) and on two GPUs (one worker thread per GPU). Unfortunately, we failed to use `nvprof` on `DGX-1` with larger number of GPUs due to technical reasons. Anyway, it was found that CUDA memory management functions `cudaMalloc`, `cudaFree` and `cudaMemcpy` take about 70–80% of overall rendering time, which can be the reason of such poor scalability.

The amount of onboard memory per GPU (see Table 2) is in most cases much larger than memory required for one worker thread. Therefore one can try running multiple worker threads per GPU. The results of such tests are presented in Fig 9. Diagrams contain rendering durations for each problem on each machine. Only one GPU is used on each machine, but there are 1, 2, or 4 worker threads per GPU. As one can see, there is a scalability issue. Almost all results do not show any speedup of the rendering when the number of workers grows. The only appropriate case is level 3 on `DGX-1`. This problem is subject for a future investigation.

## 3.2 Visualization quality

Examples of images rendered for datasets `hump` and `cascade` are shown in Fig 10. Each of those high quality images is rendered in hundreds of milliseconds on `DGX-1` and `Tesla`, and in about a second on `GeForce`.

ReVisE implements progressive rendering to provide better interactivity. When visualizer controller input changes due to a user action, a new image is rendered at level 0 and sent to the client; if at that moment the input is still the same, an image is rendered at level 1 and sent to the client, and so on, until all metadata layers are rendered. At user action, any rendering of levels 1 and higher in progress is cancelled, and the process starts again from level 0. As a result, in the progressive rendering mode user sees lower quality images first, followed by higher quality images as soon as they become available. Fig 11 presents examples of images rendered at different levels and demonstrates that, on the one hand, lower-quality images still carry useful information, and, on the other hand, higher-quality images provide more detailed view of visualized fields' features.

To further assess the quality of images generated by ReVisE, similar images have been also obtained with ParaView. To make it possible, a utility has been created (and added to the ReVisE repository) that converts files from any format readable by ReVisE to the CGNS format readable by ParaView. Fig 12 shows ParaView output on the left, and ReVisE output on the right. Notice that ParaView images have been generated using the original 'cascade' mesh data, without any intermediate approximation; therefore, ParaView images should be interpreted as "ground truth".

Comparing ParaView and ReVisE images, one can conclude that the corresponding images are very close to each other, although not identical. The sources of differences are (1) small differences in color transfer function and probably different color transformations in rendering

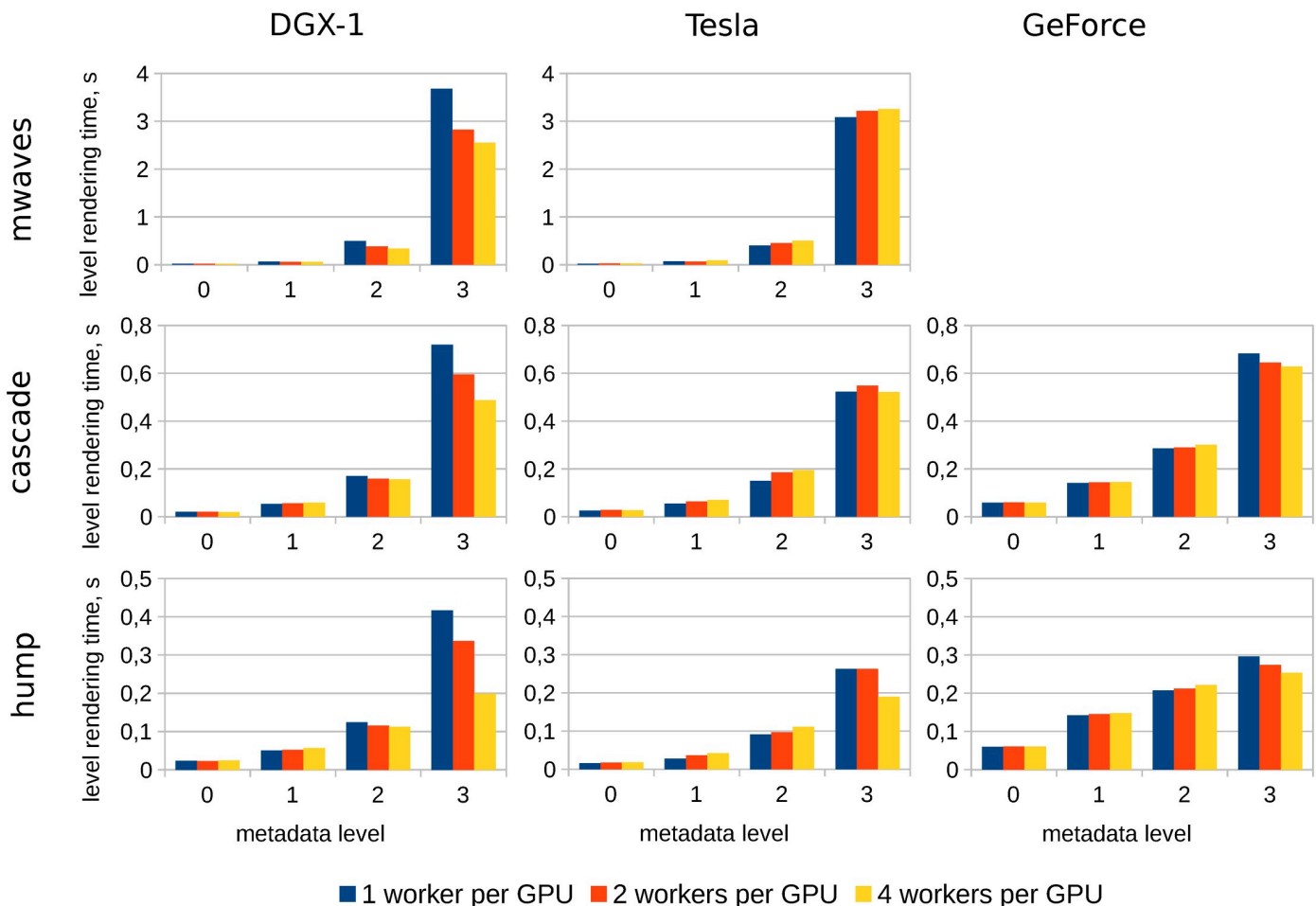

**Fig 9. Dependency of rendering duration on the number of workers per GPU.** The rendering is performed on one GPU of each machine, so all worker threads use one GPU. The rendering algorithm is maximal intensity projection. The combination `GeForce-mwaves` is absent because it could not be processed due to insufficient memory—this issue is to be resolved in further versions.

pipelines of ParaView and ReVisE, (2) different light source positions, in the case of isosurfaces, (3) small differences in camera positions, (4) different input data (original mesh for ParaView and preprocessed blocks for ReVisE). Let us notice that the comparison of zoomed isosurface parts in Fig 12 proves that the rendering quality in the presented example is sufficient, and as the detail level increases, the geometry of isosurfaces obtained with ReVisE converges to "ground truth" (notice that the images C, D at level 3 in Fig 11 are the same as D in Fig 12).

## 4 Discussion

The ReVisE system is designed with interactive visualization in mind. To reach this goal, all visualization-specific data processing has been split into two phases, which are the preprocessing and the visualization-time processing. The preprocessing is done once per problem. It contains computationally intensive operations that are done offline, i.e., without user interaction. Further visualization of the resulting dataset does not involve such lengthy operations and can be done in real time. Quality of real-time visualization is determined by the effective 3D texture resolution, which in turn depends on computational resources available at visualization

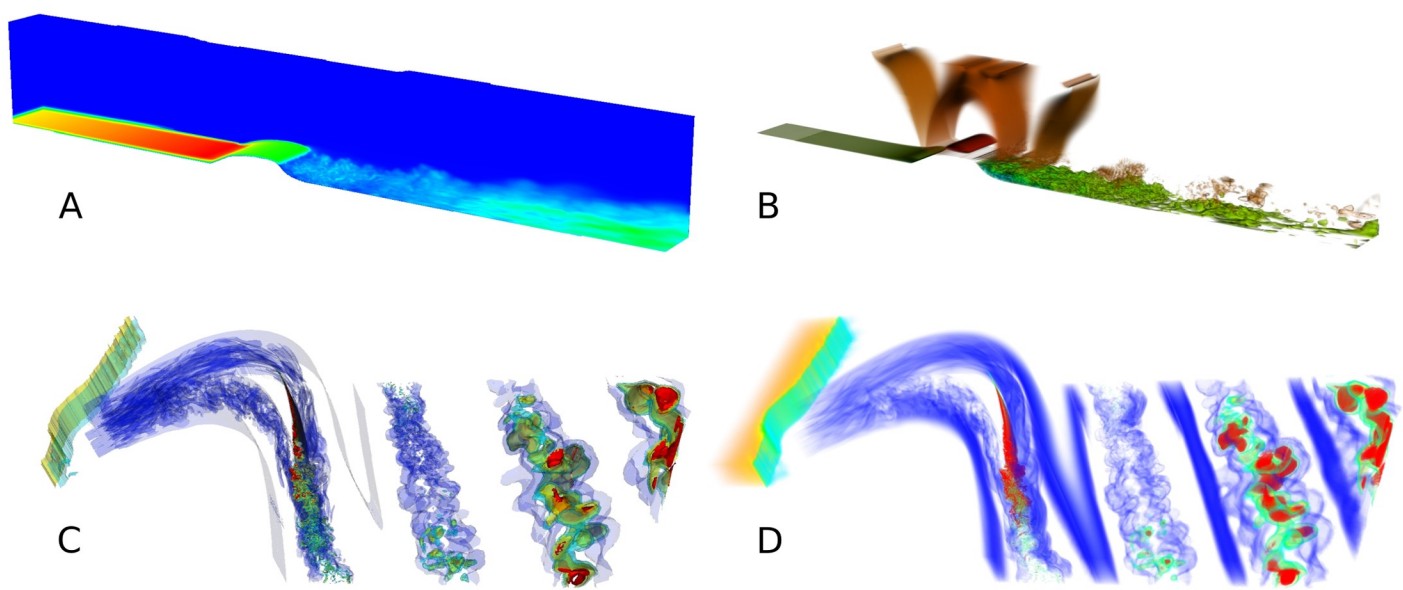

**Fig 10. Examples of images rendered by ReVisE.** All images are rendered at level 3 with block depth 8, which corresponds to 3D texture resolution 2049 in each dimension. A: `hump` dataset, maximum intensity projection. B: `hump` dataset, colormap with transparency and per-sample lighting. C: `cascade` dataset, multiple transparent isosurfaces. D: `cascade` dataset, colormap with transparency.

time. As has been shown, medium-quality images can be obtained in real time on a desktop system equipped with a GeForce class GPU.

## 4.1 Comparison with other systems

The most recognized visualization system for numerical simulation results is ParaView [2]. Therefore, it is natural to compare visualization performance of ReVisE with that of ParaView.

Preprocessing stage is done offline in ReVisE, resulting in a persistent dataset for further visualization. Therefore, the preprocessing does not affect user experience during visualization session. At visualization time, ReVisE performance does not strongly depend on the size of the original dataset, because data size at each metadata level is limited. Practically, larger original datasets result in more detailed octrees generated at the preprocessing stage, but the total number of levels sufficient for visualization can be limited explicitly, and the worst possible performance for different levels of a dense dataset is known—see `mwave` example above. In contrast to that, ParaView processes the original dataset at each startup, forcing user to wait some initial time that can be quite large and depends on dataset size.

To obtain reasonable animation performance with ParaView, a specific format of the dataset is required: ParaView volume rendering algorithm works fast only in the case of the "image" data format, which corresponds to a regular Cartesian grid. Therefore, data has to be preprocessed (resampled to image) before animation—trying to preprocess on-the-fly leads to prohibitively low animation frame rates (each time step requires many seconds to render), which is a consequence of having to process the original dataset.

In order to compare the performance of ReVisE and ParaView, the `hump` model has been chosen. The numerical solution contains 100 time frames of unsteady gas flow, therefore, the speed of time history animation can be compared. That has been done on the `Tesla` system described in Table 2. The original dataset has been converted into the `pvti` image format for visualization in ParaView. Each time frame has been represented by regular grid 1024 × 152 ×

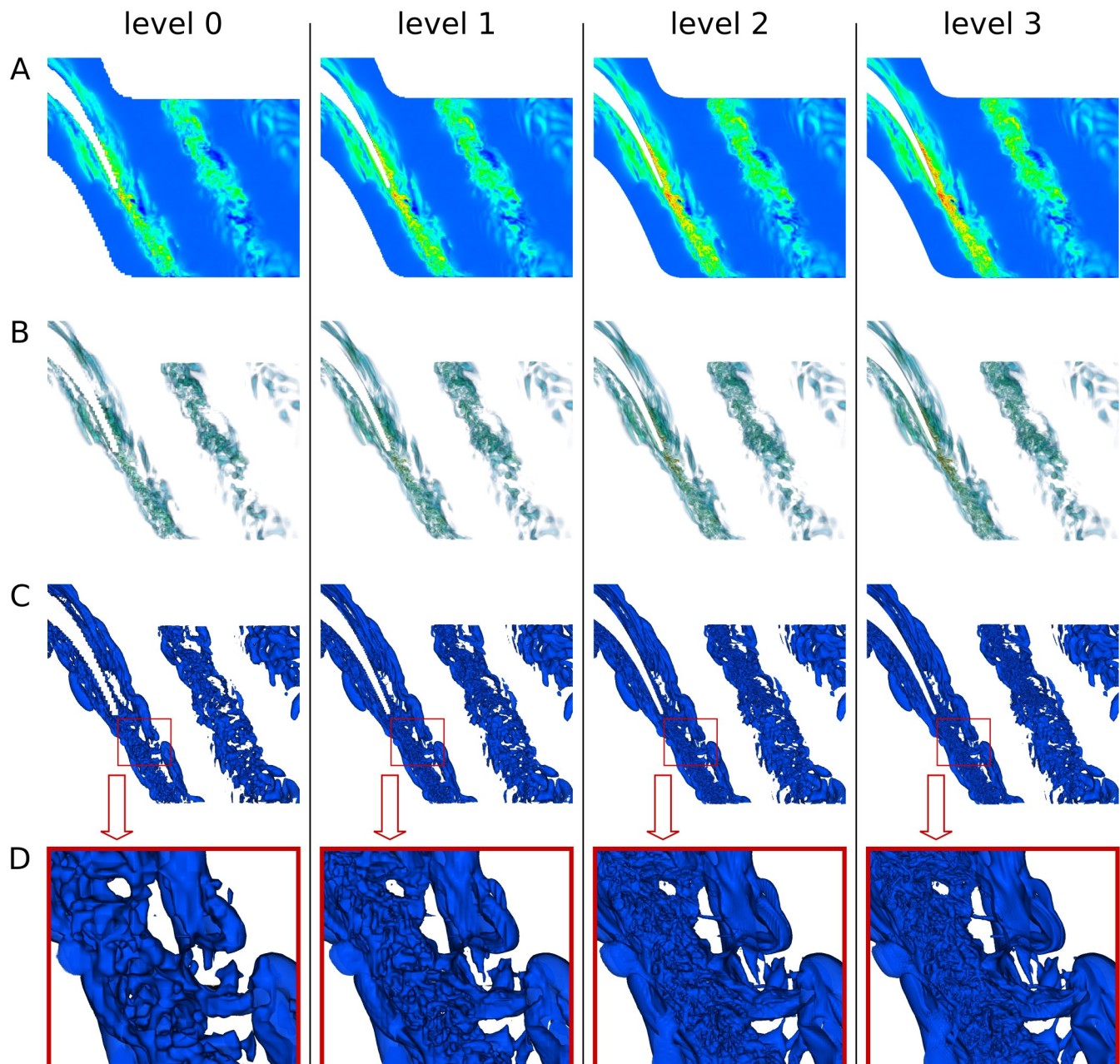

**Fig 11. Images rendered for the `cascade` dataset at different levels.** All images are rendered at block depth 8. A: Maximum intensity projection. B: Colormap with transparency and per-sample lighting. C: Single opaque isosurface. D: Single opaque isosurface, zoomed.

67. This size of the grid allows to compare visualization speed in ParaView against visualization speed of level 2 in ReVisE: the s3dmm dataset of level 2 with depth 8 is expanded to the dense grid with 1024 cells per edge. Therefore, dense grids in ParaView and ReVisE visualization datasets are expected to be equal.

Both ParaView and ReVisE were running on Tesla in server mode, and the corresponding client side was running on a remote PC connected to server via Internet.

As a preliminary test, interactive pan/zoom/rotate operations have been tested in ParaView and ReVisE, without animation along the time axis. Volume rendering speed for ParaView

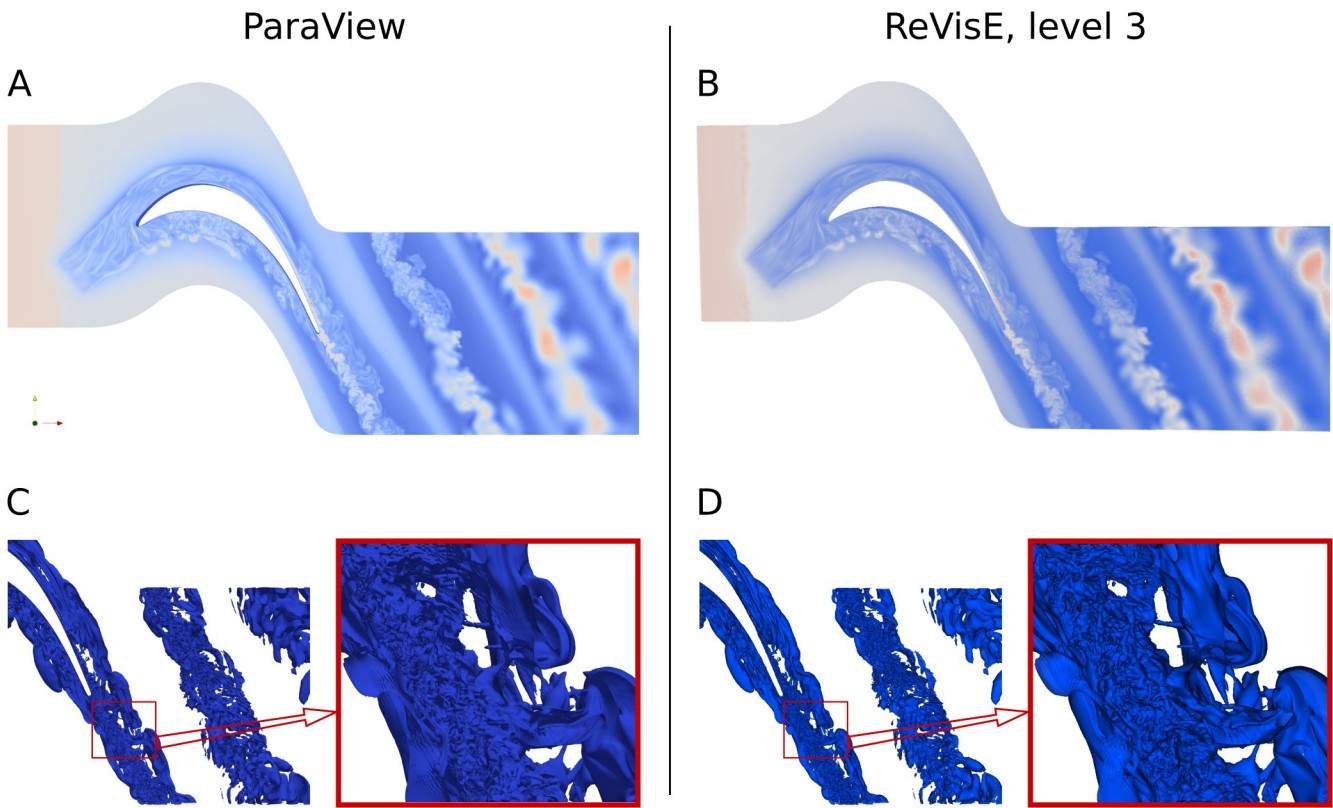

**Fig 12. Images rendered for the `cascade` dataset by ParaView (A, C) and ReVisE at level 3 (B, D).** A, B: Field at the domain boundary surface (colormap). C, D: Single opaque isosurface of another field; the same isosurface zoomed.

and ReVisE during those operations is similar when one GPU is used (we were not able to use more than one GPU in ParaView without installing any plugins). Specifically for the hump dataset, ParaView reports frame rates between 5 and 10, and it can be seen that a simplified rendering algorithm is used while the user holds the mouse button. In ReVisE, frame rates are higher (about 20 fps), but only when the progressive rendering option is enabled. This provides a bit smoother interaction experience. When the progressive rendering option is disabled, ReVisE renders 4–5 frames per second at level 2 and 1–2 fps at level 3. With larger number of GPUs, ReVisE frame rates are higher, e.g. with two GPUs it is about 10 fps at level 2 and 4–5 fps at level 3.

As the main test, the total animation duration of all 100 time frames has been measured. Volume rendering mode has been used in the test. The results are presented in Table 4. The second column represents results for ParaView: all 100 frames were played in 828 seconds, which corresponds to frame rate of 0.12 fps. Next two columns represent the same results for

**Table 4. Comparison of animation duration with ParaView and ReVisE.** Notation 2/4 means that 2 GPUs are used with 4 worker threads per GPU; 1/1 means that 1 GPU with 1 worker thread is used.

|  | ParaView | ReVisE | | |
|---|---|---|---|---|
|  |  | 2/4 | 1/1 | preprocessing |
| **duration, s** | 828 | 24 | 38 | 583 |
| **frame rate, fps** | 0.12 | 4.17 | 2.63 |  |

ReVisE. Column 3 contains results for the case when 2 GPUs are used for rendering with 4 worker threads per GPU (notation is 2/4). Column 4 corresponds to the case of sequential rendering with only 1 worker thread and 1 GPU used (notation 1/1).

First of all, the measurement with `nvidia-smi` has shown that ParaView does not load GPU more than 35% and volume rendering is performed very fast.

Possible reasons for such poor animation frame rate in ParaView are low speed of reading data from HDD and memory allocation/freeing due to VTK objects creation. It is worth noticing that pan/zoom/rotate operations are performed in real time, because they involve volume rendering only and do not involve loading from HDD or VTK objects allocations.

Therefore, one can see that ReVisE does the animation significantly faster. In our opinion, the main reason for that is the use of sparse visualization dataset, which becomes dense only in GPUs memory.

The last column of Table 4 contains the duration of visualization dataset generation (preprocessing). As one can see, that time is even smaller than the duration of animation playing in ParaView. It should be noticed that the ReVisE visualization dataset is only generated once and further reused multiple times. Also it should be noticed that the generation of `pvti` files for animation in ParaView requires much more time than the ReVisE preprocessing: all 100 frames were generated in almost 4 hours. The sizes of these datasets are almost equal: `pvti` dataset is about 60 Gb, whereas s3dmm dataset is 70 Gb. It is worth noticing that s3dmm dataset contains levels 0–3, which is redundant for the comparison, since, as already mentioned, ParaView animation is compared against ReVisE animation at level 2. A much smaller s3dmm dataset could be generated for level 2.

In our opinion, the big difference in animation speed between ReVisE and ParaView is mostly due to non-optimal data flow in ParaView. ReVisE has been specifically designed to be efficient in that respect from day 0, whereas in the case of ParaView the main idea was the flexibility and potentially a wide range of general functionality. ReVisE dataset storage format ensures the ability to read data contiguously, without having to perform many seek operations. The amount of data to be read varies, depending on the desired detail level, which allows to generate preliminary low-resolution image very fast, but still have high data read speed for higher resolution. Once read, ReVisE sparse data needs to be uploaded to GPUs block-by-block, and processed on the GPUs. In the case of ParaView, it may happen that data reading operations are not as efficient, and the data undergoes processing on CPU after loading, which may take considerable time.

The use of `pvti` image files in ParaView has one more problem. While this approach works for medium-size datasets, it does not scale well, because ParaView volume rendering algorithm takes a 3D image and uploads it to the GPU as a whole. Firstly, the entire image may not fit into GPU memory if it is large enough. Secondly, uploading the entire image to GPU takes considerable time; if the original dataset consists of fields on a non-homogeneous mesh, it is more efficient to adopt our approach and upload sparse fields to GPU and then convert those fields into dense ones directly on GPU.

Another option for ParaView is using a plugin, such as OSPRay or NVIDIA IndeX. The work [26] presents the results of using these two plugins, as well as the built-in volume rendering functionality, to animate a huge dataset with mesh containing 6912×3456×384 (about $9 \cdot 10^9$) nodes. It is shown that with NVIDIA IndeX, animation frames are generated at 1.48 fps on 8 cluster nodes, each having one GPU; before animation starts rendering, the startup time of 49 s is necessary.

There are other systems for interactive visualization. FAST is an open-source system that provides algorithms to visualize dataset from the medical domain. According to [7], the system reaches good rendering times for CT images (about 2 seconds for 512×512×426 image);

however, FAST seem to be using only dense data, while for many datasets with irregular geometry and non-homogeneous unstructured grids, taking data sparsity into account gives significant performance gain. Also, we did not find any published results on animation performance.

Sight [1] is a system used to render animated particle data and claimed to reach frame rates of 30 fps and more on DGX-1 for datasets of up to 600 million particles. The system uses OSPRay and NVIDIA OptiX for rendering using both CPU and GPU. We did not find additional details about the Sight system and its performance; it seems to be a closed-source project used internally at ORNL.

## 4.2 Further work

According to the results of ReVisE rendering performance measurement, we conclude that the scalability across multiple GPUs of a single computational node is far from ideal. Profiling data show that some operations take longer time when additional GPUs are used. Those operations are memory allocations on GPU and data transfers from host to GPU memory. The duration of these two kinds of operations significantly impacts scalability, which we did not expect. Current design of the rendering code involves quite large number of memory allocations at each rendering operation. We plan to significantly improve this situation by changing the source code such that GPU memory is reused. As a result, rendering times will be less, and the scalability across multiple GPUs is expected to be better. However, the problem of slowing down data transfers when multiple GPUs are used still has to be addressed and needs further investigation. In general, the problem of slow data transfers is partly solved by employing latency hiding; as we have shown, this may work even within a naive attempt to just increase the number of worker threads per GPU; a more elaborate design is expected to give better results. Another idea is distributing the rendering across different computational nodes, each having one GPU (as in [26]). Our further plans include the implementation of this approach to parallelize the rendering on a cluster. One more way to address the problem is to reduce data transfers using a fast compression algorithm, such as [27, 28].

Currently the ReVisE project has reached the proof-of-concept state, demonstrating the feasibility of our approach to the visualization of large time-dependent datasets. Further plans include enhancements of performance, usability, and stability necessary for production usage. Performance and scalability improvements include the development of remote execution components allowing to parallelize visualization server across nodes of a cluster (this is currently work in progress). For further scaling to larger datasets, preprocessing needs to be augmented with the ability to merge octrees and fields obtained separately for subdomains. Another way to enhance visualization performance is to implement a mechanism similar to ray-guided rendering, as proposed in [9], in order to reduce the number of blocks to be rendered. An important feature to implement is the API for in-situ visualization, with ReVisE visualization dataset obtained as the output. One attractive direction of further evolution is the integration with ParaView using its plugin mechanism.

## 5 Conclusion

A new system, ReVisE, for the remote visualization of time-dependent scalar fields defined on large meshes, has been designed and implemented. The design combines a number of known ideas, such as using octrees, and adds new ones, like efficient data structure to store and use octrees. A number of real-life CFD problems have been considered; tests show that ReVisE outperforms other systems in some cases, especially for large time-dependent datasets, due to its specific design. Nevertheless, there is a big potential for further improvement of performance, especially by optimizing GPU memory operations and parallelizing across several

nodes of a cluster. ReVisE has a limited functionality, and this can be addressed in a number of ways, one of which is integration with ParaView.

## Acknowledgments

Authors are thankful to Supercomputer Center "Polytechnic" of Peter the Great St. Petersburg Polytechnic University for providing access to a DGX-1 machine and the "Tornado" supercomputer.

In memory of professor Nikolay Shabrov, who launched the ReVisE project.

## Author Contributions

**Conceptualization:** Stepan Orlov, Alexey Kuzin, Andrey Pyatlin.

**Data curation:** Vladislav Kiev.

**Formal analysis:** Alexey Kuzin, Vladislav Kiev, Andrey Pyatlin.

**Investigation:** Alexey Kuzin, Alexey Zhuravlev, Vyacheslav Reshetnikov, Vladislav Kiev, Andrey Pyatlin.

**Methodology:** Stepan Orlov.

**Project administration:** Stepan Orlov.

**Resources:** Vladislav Kiev, Andrey Pyatlin.

**Software:** Stepan Orlov, Alexey Kuzin, Alexey Zhuravlev, Vyacheslav Reshetnikov, Egor Usik, Vladislav Kiev.

**Validation:** Alexey Kuzin, Egor Usik, Vladislav Kiev.

**Visualization:** Alexey Zhuravlev.

**Writing – original draft:** Stepan Orlov, Alexey Kuzin, Alexey Zhuravlev.

**Writing – review & editing:** Stepan Orlov, Alexey Kuzin, Vyacheslav Reshetnikov, Egor Usik, Vladislav Kiev, Andrey Pyatlin.

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
