## [Decision Letter · Decision Letter 0]

31 Dec 2020

PONE-D-20-33961

ReVisE: Remote visualization environment for large datasets

PLOS ONE

Dear Dr. Orlov,

Thank you for submitting your manuscript to PLOS ONE. After careful consideration, we feel that it has merit but does not fully meet PLOS ONE’s publication criteria as it currently stands. Therefore, we invite you to submit a revised version of the manuscript that addresses the points raised during the review process.

We look forward to receiving your revised manuscript.

Kind regards,

Frederique Lisacek

Academic Editor

PLOS ONE

Journal Requirements:

3. We noted in your submission details that a portion of your manuscript may have been presented or published elsewhere:

'Part of this work is to be published in "Communications in Computer and Information Science" at the end of December, 2020. The paper is titled "Core Algorithms of Sparse 3D Mipmapping Visualization Technology" and is further referred to as s3dmm. Its preprint will be uploaded along with other files. There is some intersection of materials published in that paper with the manuscript being submitted (and is further referred to as ReVisE). Those intersections are as follows.

The "Preprocessing Pipeline" section of s3dmm has common parts with the "Sparse 3D mipmapping" subsection of ReVisE. In particular, Fig 1 of s3dmm coincides with Fig 4 of ReVisE. But there is no one-to-one correspondence between those two sections, because we focused on different things. In ReVisE, we present a reduced version from s3dmm, but the new version also reflects our experience gained during the months after the s3dmm paper was prepared.

The "Octree storage optimization" section of ReVisE is close to sections "Octree Generation" and "Optimized Storage of Octree Data Structure". Still we decided to include that section into ReVisE because we find the proposed octree compression algorithm quite interesting and useful.

However, we do not consider ReVisE as a duplicate of s3dmm. The differences are that

- s3dmm presents the core technology in more details, compared to ReVisE.

- The results presented in s3dmm do not cover the visualization system as a whole: we write about the rendering of just one block on a PC equipped with GPU, and that was what we had at the moment of writing.

- On the other hand, ReVisE presents results of performance tests on a number of architectures (DGX-1, Tesla, GeForce), involving remote operation via web application, using video streaming service, and progressive rendering.

- ReVisE is focused on the presentation of the new visualization system as a whole thing, while s3dmm is dedicated to core algorithms.'

Please clarify whether this publication was peer-reviewed and formally published.

If this work was previously peer-reviewed and published, in the cover letter please provide the reason that this work does not constitute dual publication and should be included in the current manuscript.

4. Please amend the manuscript submission data (via Edit Submission) to include authors Alexey Kuzin, Alexey Zhuravlev, Vyacheslav Reshetnikov, Egor Usik, Vladislav Kiev and Andrey Pyatlin.

5. Please ensure that you refer to Figure 4 in your text as, if accepted, production will need this reference to link the reader to the figure.

Reviewers' comments:

Reviewer's Responses to Questions

**Comments to the Author**

1. Is the manuscript technically sound, and do the data support the conclusions?

Reviewer #1: Yes

Reviewer #2: Partly

2. Has the statistical analysis been performed appropriately and rigorously? 

Reviewer #1: N/A

Reviewer #2: N/A

3. Have the authors made all data underlying the findings in their manuscript fully available?

Reviewer #1: Yes

Reviewer #2: No

4. Is the manuscript presented in an intelligible fashion and written in standard English?

Reviewer #1: Yes

Reviewer #2: Yes

5. Review Comments to the Author

Reviewer #1: In this manuscript the authors present ReVisE, a new software for interactive visualization of large datasets from complex simulations. The manuscript is well written with a clear introduction to the problem, results and discussions. The design of the system is well described, including data structures, algorithms and software architecture. The performance of their software is evaluated with various datasets.

A major concern is the availability of the software. It is stated that the source code will be made available at GitHub as soon as the paper is published. However, as it is increasingly acknowledged by the scientific community, the source code of any scientific software funded by public institutions and aimed at open source should be available for review. Besides, having an instance with a simple visualization example in a server as demo with a link provided to the reviewers (and later for potential users would also be beneficial) will make a stronger publication.

Regarding technical details of implementation and deployment, it would be useful to add in Fig. 8 the programming languages of the main software components. Other helpful additions could be mentioning what are the dependencies if there are any and an estimate of the percentage of new code that was implemented or the one re-used (the authors mentioned that it is based on source code of OpenGL fragment shaders found in the Visualization Library).

The authors claim that tests show that ReVisE outperforms other systems in some cases, especially for large time-dependent datasets. However, the comparison with other systems does not seem quantitative. Even if numbers are from the respective publications of the other tools, a comparison table indicating dataset, hardware, execution time and some comments specific to each system would be preferred.

Minor suggestions are made to improve the manuscript:

• Results: please add details of the deployment used for the tests. Where all servers in the same computer?

• Line 534: GPU memoy -> GPU memory

• Line 550: didn’t -> did not (too informal otherwise)

Reviewer #2: The paper introduces a new visualization system, called ReVisE, that aims at remotely visualizing large---not of the reviewer numerical simulation---datasets. I have appreciated to review this article, that has an overall interesting content, and that takes the pre-processing approach in order to solve a task that is otherwise not reachable at the scale of nodes the author intend to work with in real time. We have nonetheless to wait line 8 to understand the kind of data that are manipulated, also something like "for large numerical simulation datasets" in the title and abstract would immediately reduce the scope of the article, as the purpose is not to visualize other kind of large datasets such as corpus of texts.

In the introduction, the lack of available visualization software that scales up is emphasized (l.13) to motivate the introduction of their system or with performance up to 10^6 nodes. If the scaling issue is tackled clearly in the rest of the article, does the proposed platform intended to be available Open Source, is there a Demo somewhere or at least a video of the platform?

The related work relies more on other softwares / products, than on what are the existing technologies, what is the state of the art this Research article is linked to.

In order to help relates this article to existing work, references such as BEYER 2014 State of the Art in GPU Based Large Scale Volume Visualization or Wang 2020 Portable interactive visualization of large-scale simulations in geotechnical engineering using Unity3D, which introduces not only the different softwares but also the different technologies to motivate deeply their approach, could be used. A beginning of argumentation and a clear introduction of the plan should be made at the end of the introduction.

Sections, paragraphs and subparagraphs at least should be numbered, as it would really simplify the reading, especially when forward references are made as the size of the police is not sufficient to make a clear distinction between sub-paragraphs and sub-sub-paragraphs. When pointing to Forward or Backward reference in your article, you should refer precisely.

The task to be solved should be clearly stated as such in the introduction and related to the research question the author want to solve; formulating it properly, would induce the steps to achieve to properly evaluate the proposed platform, in comparison with other existing platforms.

The article appears to be an extension of the already published article [12] (from line 66 to line 315). It should be stated as such both in the abstract and in the introduction. The new contributions of this article have to be clearly stated from the introduction. To help clarify this point, the paragraph Sparse 3D mipmapping, should not only said that the sparse mipmapping has been proposed in [12] but also mention that the whole paragraph reintroduce the concepts for the sake of clarity. Having an Arxiv version of [12] would definitely help the interested reader. When full paragraphs (l.224 to 248) are taken from [12] they should be quoted and referred as such. Figures from [12] (Fig.3 that corresponds to Fig. 2 and 5 of [12], Fig 4 that is Fig 1 of [12], Fig 5 that is Fig.7 of [12]) cannot be reproduced as such for evident copyright reasons: they should either differ or be cited from [12]: in this case, authors should have the consent of Springer for reusing them in another article.

L.242 to 270:

How to compare two things that will not be comparable, with a part of pre-processing with the ReVisE and other system that are synchronous? The only way is to compare at least the full process and show what are the gains / costs of making it asynchronous. The other point, is that the scalar field has to change over time, also how are you going to take that into account if it is in the part of the pre-processing? This paragraph lets understand there is still work to do on that (l.269-270), which would explain that the pre-processing is not taken into account (l.244-245); and in this case how do you evaluate a reasonable duration (l.245)? How do you compare it to existing frameworks? l. 266 to 268: are there corroborating results to assert the advantage of the proposed algorithm as [12] does not show any baseline for comparing?

Moreover the algorithm for stage (5) should be given.

The new contributions of the article comes from line 316 and forward, include the introduction of the general architecture of the visualization system, moving to the engineering part of the project, mostly descriptive of the implementation; the level of details is largely sufficient to follow.

Results part (and I would suggest to rename this part Results and Evaluation):

To evaluate the performances of the proposed system, the proposed baseline in Comparison with other systems relies mainly on open systems, and particularly Paraview. As it is mentioned Paraview does the pre-processing on the fly, so it is going to be hard to compare two processes that do not rely on the same basis. In order to keep this baseline, results on preprocessing time and visualization processing time should be gathered in one table for both systems. If they are equivalent for the different datasets, then we can conclude that the one that has the pre-processing is going to outperform the one that has to recompute each time. A Table with figures for each stage is expected at that level for at least ReVisE and the baseline, for each proposed dataset and for the same level. Additionaly, l. 403 (preprocessing times have also been measured) contradicts l. 245, and no figures are given.

In the Visualization performance (first occurrence l.428, as a second paragraph l.557 is named the same) the results are centered on the presented system; some tables presenting the datasets with the number of nodes at each level would be an asset (l.435-440). The approach is interesting. For helping the reading l.465 to 475 should be put before l.457 to 464 in order to avoid a forward reference on l.456.

In the Visualisation quality, there is the need to evaluate it properly: which task has to be solved? which metric is used to show the improvement on the quality and how do we compare this quality to a baseline (what would have been generated by Paraview)? Only experts of the field could answer about the quality of such visualization and respond to what is asserted in lines 487 to 490, as in fact to restrain to level 3 might not be as assumed in the introduction the level of quality the experts would need to have. This evaluation should be done by mixing those visualization with visualization generated with Paraview.

In the paragraph Comparison with other systems, you mention l.504 "According to our experience" for the better interactivity: this is very subjective and need to be quantified. Particularly, I would have expected to see results on the number of fps rendered with Revise compared to other systems. Once again a well designed Table with different criteria for comparing the systems would be welcome.

I would put the second paragraph Visualization performance in Future work (more than Further work) as from line 564 it concerns only Future work.

A last question has to be raised with the final sentence: why would not have started by integrating the ReVisE in the Paraview? I think a different formulation can be found for this point.

Additional remarks:

Direct style (for instance l.60-61), imperative forms (l.110, l.281 for instance) should be avoided in a scientific article. Statements should be accompanied by proovable arguments; particularly when coming to evaluation this is definitely needed. I would have expected the visualization performance to be compared to existing softwares, and particularly to Paraview as far as the number of nodes are comparable (maybe for instance with the hump dataset). It would have included in this case the preprocessing stage and the visualization stage, as it has been partially done in [12], showing the gain that we can expect from the new platform. Coming back to the assertion made on line 60-61 there is no proof (i.e. no comparison on an A/B testing for instance) of the visualization made. A baseline would be again to simplify the problem in a first instance to make two comparable visualization one obtained by Paraview for instance and one obtained by your method, and then scaling, and showing what happens. A video comparing the two could also be an asset. A particular attention to articles (a and the) should be given, particularly when they are missing (for instance l.339: it uploads the image to the shared memory; l.343: the video streaming service checks the shared memory cotents). Bullets should be avoided in an article (it is not a presentation); a table is often a good replacement (l 405 to 437 and l.420 to 427)

Some remarks on Figures:

All the figures come at the end of the article and are not included in proper way in the article, which makes things difficult to follow.

Figure 1: I would let the mesh behind in B (as it is done in C). Moreover, the frame used for the quadtree should be marked as such in A, just to be sure that all Figures from A to F are at the same dimension of representation (which does not seem to be the case).

Figure 2: I think the mesh chosen for illustrating a bit hard to follow and would let it by transparency under the different quadtrees generated.

Figure 3: I would add colors to help following the octree, or / and draw the graph tree corresponding.

Figure 4: having the mesh by transparency under the quadtrees generated would help.

Figure 5: It differs only from one node from Springer. A legend indicating nodes signification directly on the Figure it self would help (and not in the Legend below the Figure). The Legend below figure 5: the second B should be C.

Figure 8: two inversions: Numerical solution (oi inverted) and Web server: visualization controller (rt inverted)

Figure 9: ReVisE web interface and not ReVesE in the legend

Suggestions of figures:

A figure similar to Figure 3 of [12] would clarify lines 126-127.

A figure might help to clarify l.154 - 162.

Mispelling / other remarks:

l.403: preformance instead of performance

l.345: what information does the sentence bring?

The article would really benefit to wait for further results that are underwork (particularly the pre-processing stage needed to make full comparison) and to be evaluated using a strong baseline to compare to other existing frameworks / existing methods; this is needed to show that there is a real improvement and not an artefact of improvement, by just lowering the quality of the visualization given when the dataset becomes to big. I let it nonetheless in major revisions if those strong objections can be lifted previously to any final acceptation, considering the work of the authors.

6. PLOS authors have the option to publish the peer review history of their article (what does this mean?). If published, this will include your full peer review and any attached files.

Reviewer #1: **Yes: **Aivett Bilbao

Reviewer #2: No

---

## [Author Response · Author response to Decision Letter 0]

11 Apr 2021

1

To Editors

1.1

Q: Please ensure that your manuscript meets PLOS ONE’s style requirements,

including those for file naming. The PLOS ONE style templates can be found

at...

A: Done

1.2

Q: We note that you have stated that you will provide repository information

for your data at acceptance. Should your manuscript be accepted for publica-

tion, we will hold it until you provide the relevant accession numbers or DOIs

necessary to access your data. If you wish to make changes to your Data Avail-

ability statement, please describe these changes in your cover letter and we will

update your Data Availability statement to reflect the information you provide.

A: Yes, we have made all the data accessible. First of all, we have uploaded

Revise’s sources on Github: https://github.com/deadmorous/revise. It is

distributed under AGPL-3.0 license. Also we have deposited all input data into

protocols.io. Corresponding DOI are:

• Build ReVisE on Ubuntu:

http://dx.doi.org/10.17504/protocols.io.bruwm6xe

• Prepare and run test on available dataset:

http://dx.doi.org/10.17504/protocols.io.bruzm6x6

1.3

Q: We noted in your submission details that a portion of your manuscript may

have been presented or published elsewhere: ...

Please clarify whether this publication was peer-reviewed and formally pub-

lished.

If this work was previously peer-reviewed and published, in the cover letter

please provide the reason that this work does not constitute dual publication

and should be included in the current manuscript.

A: Yes, the article ”Core Algorithms of Sparse 3D Mipmapping Visualization

Technology” is peer reviewed and published.

This is the reference:

Orlov S., Kuzin A., Zhuravlev A. (2020) Core Algorithms of Sparse 3D

Mipmapping Visualization Technology. In: Voevodin V., Sobolev S. (eds) Su-

percomputing. RuSCDays 2020. Communications in Computer and Informa-

tion Science, vol 1331. Springer, Cham.

https://doi.org/10.1007/978-3-030-64616-5_36

The reasons requested are in the cover letter.

1.4

Q: Please amend the manuscript submission data (via Edit Submission) to

include authors Alexey Kuzin, Alexey Zhuravlev, Vyacheslav Reshetnikov, Egor

Usik, Vladislav Kiev and Andrey Pyatlin.

A: Done

1.5

Q: Please ensure that you refer to Figure 4 in your text as, if accepted, produc-

tion will need this reference to link the reader to the figure.

A: The reference to Figure 4 inserted.

2

To Reviewer 1

2.1

Q: A major concern is the availability of the software. It is stated that the

source code will be made available at GitHub as soon as the paper is published.

However, as it is increasingly acknowledged by the scientific community, the

source code of any scientific software funded by public institutions and aimed

at open source should be available for review. Besides, having an instance with

a simple visualization example in a server as demo with a link provided to the

reviewers (and later for potential users would also be beneficial) will make a

stronger publication.

A: The source codes of Revise are now available at GitHub under AGPL-

3.0 license: https://github.com/deadmorous/revise. Also all input data is

deposited into protocols.io. Corresponding DOI are:

• Build ReVisE on Ubuntu:

http://dx.doi.org/10.17504/protocols.io.bruwm6xe

• Prepare and run test on available dataset:

http://dx.doi.org/10.17504/protocols.io.bruzm6x6

2.2

Q: Regarding technical details of implementation and deployment, it would be

useful to add in Fig. 8 the programming languages of the main software compo-

nents. Other helpful additions could be mentioning what are the dependencies

if there are any and an estimate of the percentage of new code that was imple-

mented or the one re-used (the authors mentioned that it is based on source

code of OpenGL fragment shaders found in the Visualization Library).

A: We fixed Fig. 8.

Presently Revise does not depend on any external modules, besides trivial

ones (such as Qt, googletest). It also depends on VisualizationLibrary, which

is used as external library and is loaded as Git submodule. Presently Revise is

intended to support two implementations of rendering: with CUDA and with

OpenGL. The article does not span the second one because it is not developed

well now and, probably, will not be developed in the future due to some diffi-

culties when multiple GPUs used and due to absence of advantages over CUDA

implementation. Revise depends on VisualizationLibrary when it is built in

OpenGL mode because the rendering works via VL API. The dependency on

VL in CUDA mode is rudimentary.

When we write that our implementation of rendering is based on source

code of shaders of VL we mean that we refer to algorithms used but the code is

almost fully re-written. VL rendering is OpenGL-based and written in GLSL,

when our rendering is CUDA-based so it has to be written from scratch as

CUDA kernels. The VL implementation generally contains classical algorithms,

such as volume raycasting and we used this implementation as reference. The

only thing that can be treated as some sort of copying from VL is Blinn lighting

implementation. Here is the comparison of these both sources:

Revise:

https://github.com/deadmorous/revise/blob/master/src/s3dmm_cuda/

cuda_lighting.hpp#L27

https://github.com/deadmorous/revise/blob/master/src/s3dmm_cuda/

cuda_lighting.hpp#L54

VL:

https://github.com/MicBosi/VisualizationLibrary/blob/master/data/

glsl/volume_raycast_isosurface_transp.fs#L44

https://github.com/MicBosi/VisualizationLibrary/blob/master/data/

glsl/volume_raycast_isosurface_transp.fs#L66

But it is worth noticing that Blinn lighting is well-known classical algorithm

with well-known implementation.

2.3

Q: The authors claim that tests show that ReVisE outperforms other systems

in some cases, especially for large time-dependent datasets. However, the com-

parison with other systems does not seem quantitative. Even if numbers are

from the respective publications of the other tools, a comparison table indi-

cating dataset, hardware, execution time and some comments specific to each

system would be preferred.

A: A comparison of volume rendering of ParaView and ReVisE added.

2.4

Q: Minor suggestions are made to improve the manuscript:

• Results: please add details of the deployment used for the tests. Where

all servers in the same computer? A: ReVisE can be deployed on a Linux

system by building it from the source code, or by using a docker image.

See https://github.com/deadmorous/revise#installing-revise and

https://github.com/deadmorous/revise#docker. In addition, an in-

stallation procedure is described in this protocol: http://dx.doi.org/

10.17504/protocols.io.bruwm6xe

• Line 534: GPU memoy → GPU memory A: Done

• Line 550: didn’t → did not (too informal otherwise) A: Done

3

To Reviewer 2

3.1

Q: The paper introduces a new visualization system, called ReVisE, that aims at

remotely visualizing large—not of the reviewer numerical simulation—datasets.

I have appreciated to review this article, that has an overall interesting content,

and that takes the pre-processing approach in order to solve a task that is

otherwise not reachable at the scale of nodes the author intend to work with

in real time. We have nonetheless to wait line 8 to understand the kind of

data that are manipulated, also something like ”for large numerical simulation

datasets” in the title and abstract would immediately reduce the scope of the

article, as the purpose is not to visualize other kind of large datasets such as

corpus of texts.

A: Done

3.2

Q: In the introduction, the lack of available visualization software that scales

up is emphasized (l.13) to motivate the introduction of their system or with

performance up to 10 6 nodes. If the scaling issue is tackled clearly in the rest of

the article, does the proposed platform intended to be available Open Source,

is there a Demo somewhere or at least a video of the platform? The related

work relies more on other softwares / products, than on what are the existing

technologies, what is the state of the art this Research article is linked to.

In order to help relates this article to existing work, references such as BEYER

2014 State of the Art in GPU Based Large Scale Volume Visualization or Wang

2020 Portable interactive visualization of large-scale simulations in geotechnical

engineering using Unity3D, which introduces not only the different softwares

but also the different technologies to motivate deeply their approach, could be

used. A beginning of argumentation and a clear introduction of the plan should

be made at the end of the introduction.

A: Added reference to BEYER 2014; added three paragraphs at the end of

the introduction that

• introduce main ideas from BEYER 2014;

• classify ReVisE in terms of categories defined in BEYER 2014;

• outline further sections of the paper.

3.3

Q: Sections, paragraphs and subparagraphs at least should be numbered, as it

would really simplify the reading, especially when forward references are made

as the size of the police is not sufficient to make a clear distinction between sub-

paragraphs and sub-sub-paragraphs. When pointing to Forward or Backward

reference in your article, you should refer precisely.

A: We make the manuscript according to the template of the journal and,

unfortunately, it does not have sections numbering. We added referencies to the

sections, where needed.

3.4

Q: The task to be solved should be clearly stated as such in the introduction

and related to the research question the author want to solve; formulating it

properly, would induce the steps to achieve to properly evaluate the proposed

platform, in comparison with other existing platforms.

A: In order to address and resolve those issues, we have tried to improve

problem formulation in the “Introduction” section. We also added more com-

parison results in the “Results and Evaluation” section.

3.5

Q: The article appears to be an extension of the already published article [12]

(from line 66 to line 315). It should be stated as such both in the abstract and

in the introduction. The new contributions of this article have to be clearly

stated from the introduction. To help clarify this point, the paragraph Sparse

3D mipmapping, should not only said that the sparse mipmapping has been

proposed in [12] but also mention that the whole paragraph reintroduce the

concepts for the sake of clarity. Having an Arxiv version of [12] would definitely

help the interested reader. When full paragraphs (l.224 to 248) are taken from

[12] they should be quoted and referred as such. Figures from [12] (Fig.3 that

corresponds to Fig. 2 and 5 of [12], Fig 4 that is Fig 1 of [12], Fig 5 that is

Fig.7 of [12]) cannot be reproduced as such for evident copyright reasons: they

should either differ or be cited from [12]: in this case, authors should have the

consent of Springer for reusing them in another article.

A: The section Sparse 3D mipmapping has been refactored and made shorter

with more tight references to [12]. It is stated that it retells article [12], but does

not replace, so the reader has to refer to [12] for more details. According to this

logic Figures 3, 4 and 5 are removed in order to not duplicate corresponding

figures from [12].

3.6

Q: L.242 to 270: How to compare two things that will not be comparable, with a

part of pre-processing with the ReVisE and other system that are synchronous?

The only way is to compare at least the full process and show what are the gains

/ costs of making it asynchronous. The other point, is that the scalar field has to

change over time, also how are you going to take that into account if it is in the

part of the pre-processing? This paragraph lets understand there is still work

to do on that (l.269-270), which would explain that the pre-processing is not

taken into account (l.244-245); and in this case how do you evaluate a reasonable

duration (l.245)? How do you compare it to existing frameworks? l. 266 to 268:

are there corroborating results to assert the advantage of the proposed algorithm

as [12] does not show any baseline for comparing? Moreover the algorithm for

stage (5) should be given.

A: We think, that on comparison of the speed of interactive rendering be-

tween ReVisE and other systems, we do not need to take into account prepro-

cessing time in ReVisE. One should compare the time of the rendering of the

scene: this time is important for the user because it determines the level of

convenience of usage of the system for interactive visualization. Thus, this time

may serve as quantitative characteristic of convenience of use of the system.

Reduction of this time can be achieved, on the one hand, by speeding up all

stages, and also, by transferring some operations to offline preprocessing. In

ReVisE we use, first of all, the second approach.

Changing of the scalar field over the time is also allowed. Visualization

dataset consists of metadata that is common for all time frames and arrays

of sparse fields, calculated for each time frame. An evaluation of visualization

dataset is performed at the preprocessing stage.

ReVisE preprocessing times have been included into the new version of the

paper. Also, it turns out that to gain best performance with Paraview, the

original dataset has to be preprocessed as well (otherwise, the animation per-

formance is completely unacceptable). We have compared preprocessing times

and visualization dataset sizes for ReVisE and Paraview.

We decided not to present the algorithm for stage (5) because it has already

been given in our published paper [15]. Presenting the algorithm in this paper

would increase the intersection of paper content with content published else-

where, which is not desirable. As follows from the algorithm, it is a single-pass

one, and should scale good enough as dataset size grows; of course, the com-

parison with other algorithms needs to be done and will be done as soon as we

implement our algorithm and present it in a future paper.

3.7

Q: The new contributions of the article comes from line 316 and forward, include

the introduction of the general architecture of the visualization system, moving

to the engineering part of the project, mostly descriptive of the implementation;

the level of details is largely sufficient to follow.

A: Does not require answer

3.8

Q: Results part (and I would suggest to rename this part Results and Evalua-

tion):

To evaluate the performances of the proposed system, the proposed baseline

in Comparison with other systems relies mainly on open systems, and particu-

larly Paraview. As it is mentioned Paraview does the pre-processing on the fly,

so it is going to be hard to compare two processes that do not rely on the same

basis. In order to keep this baseline, results on preprocessing time and visualiza-

tion processing time should be gathered in one table for both systems. If they

are equivalent for the different datasets, then we can conclude that the one that

has the pre-processing is going to outperform the one that has to recompute

each time. A Table with figures for each stage is expected at that level for at

least ReVisE and the baseline, for each proposed dataset and for the same level.

Additionaly, l. 403 (preprocessing times have also been measured) contradicts

l. 245, and no figures are given.

A: We have renamed Results to Results and Evaluation.

We restyled section Discussion (subsection Comparison with other system),

therefore there is comparison of animation frame rate for ParaView and Revise.

We have added preprocessing time of Revise into the table.

First of all the time of animation is compared. As one can see an animation

in Revise has higher rate than in PraView’s one. Of course, one can add pre-

processing time to Revise’s animation, but as one can see in this case it would

be a little bit faster.

Also it should be noticed that ParaView also requires preprocessing. We

have to transform original dataset into pvti format in order to provide higher

rate in ParaView. As it has mentioned in the article, it requires about 4 hours,

which is a real preprocessing time in ParaView.

It should be noticed that this comparison relates to the playing of time

history. If one does pan/zoom/rotate in ParaView without changing time frame

the volume rendering is performed very fast.

3.9

Q: In the Visualization performance (first occurrence l.428, as a second para-

graph l.557 is named the same) the results are centered on the presented system;

some tables presenting the datasets with the number of nodes at each level would

be an asset (l.435-440). The approach is interesting.

A: The table with visualization datasets parameters added. Also the second

subsection with name Visualization performance is united with Future work.

Q: For helping the reading l.465 to 475 should be put before l.457 to 464 in

order to avoid a forward reference on l.456.

A: Corresponding paragraphs are swapped.

Q: In the Visualisation quality, there is the need to evaluate it properly:

which task has to be solved? which metric is used to show the improvement on

the quality and how do we compare this quality to a baseline (what would have

been generated by Paraview)? Only experts of the field could answer about the

quality of such visualization and respond to what is asserted in lines 487 to 490,

as in fact to restrain to level 3 might not be as assumed in the introduction the

level of quality the experts would need to have. This evaluation should be done

by mixing those visualization with visualization generated with Paraview.

A: As far as of the quality of volume rendering, let us notice first that the

best possible level of detail is achieved as soon as the texel size of the 3D texture

does not exceed the size of a pixel on the screen. Because of this, the effective

texture resulution of 2048 that we have in the cascade dataset ensures the best

possible level of detail on a typical FullHD dispay (1920 x 1080 pixels), unless

the view is zoomed. Further, it might turn to be good enough to have texel size

larger than the pixel size, if the gradient of the field being visualized is not too

high. In general, the desired level of detail in the 3D texture depends on the

problem at hand and, in particular, on mesh element size (which is respected

by the ReVisE preprocessor). We would like to emphasize that in ReVisE user

has full control on the level of detail in the visualization dataset, because the

maximal number of block level is explicitly set at the preprocessing stage, which

together with block depth determines the effective resolution of the 3D texture.

Also there is the following observation about datasets from the CFD area:

those datasets are often large due to mesh refinement at the boundary, which

is necessary for the computation; however, it is typically normal to ignore those

boundary layers during the visualization. Anyways, it is up to the user how to

choose the desired level of detail by setting the mesh refinement parameter α,

the boundary refinement parameter, and the limitation on the number of levels.

3.10

Q: In the paragraph Comparison with other systems, you mention l.504 ”Ac-

cording to our experience” for the better interactivity: this is very subjective

and need to be quantified. Particularly, I would have expected to see results on

the number of fps rendered with Revise compared to other systems. Once again

a well designed Table with different criteria for comparing the systems would

be welcome.

A: A comparison of frame rates between ParaView and ReVisE are added

for hump model.

3.11

Q: I would put the second paragraph Visualization performance in Future work

(more than Further work) as from line 564 it concerns only Future work.

A: Visualization performance subsection is moved to Future work.

3.12

Q: A last question has to be raised with the final sentence: why would not

have started by integrating the ReVisE in the Paraview? I think a different

formulation can be found for this point.

A: In the beginning of our work on the project, we tried a different approach

to the visualization. One of our students was developing a plugin for Paraview.

Lessons learned were that (a) it is quite difficult to follow all the rules of Par-

aview plugin development, because of many things to be taken into account; (b)

Paraview is a large system, and in order to extend it properly with plugins, one

needs to learn Paraview source code; (c) Paraview visualization pipeline might

have performance bottlenecks that are not known in advance, and struggling

against those bottlenecks would take up considerable time. But what we really

wanted was to concentrate on the proof-of-concept implementation of our ap-

proach, rather than to learn the details of Paraview implementation. To achieve

that goal, it was significantly simpler to start from scratch and have full control

over literally any aspect of the system, than to try extending Paraview and over-

come unexpected problems we are faced to. And only after our approach proves

feasibility, it worth trying to build it into Paraview; in this case, the existing

ReVisE implementation serves as a reference for performance estimation and

helps to control that the Paraview implementation is being done properly.

3.13

Q: Additional remarks: (a lot)

Direct style (for instance l.60-61), imperative forms (l.110, l.281 for instance)

should be avoided in a scientific article.

A: Fixed

Statements should be accompanied by proovable arguments; particularly

when coming to evaluation this is definitely needed. I would have expected the

visualization performance to be compared to existing softwares, and particularly

to Paraview as far as the number of nodes are comparable (maybe for instance

with the hump dataset). It would have included in this case the preprocessing

stage and the visualization stage, as it has been partially done in [12], show-

ing the gain that we can expect from the new platform. Coming back to the

assertion made on line 60-61 there is no proof (i.e. no comparison on an A/B

testing for instance) of the visualization made. A baseline would be again to

simplify the problem in a first instance to make two comparable visualization

one obtained by Paraview for instance and one obtained by your method, and

then scaling, and showing what happens. A video comparing the two could also

be an asset.

A: We added comparison of animation rate in ParaView and ReVisE. The

preprocessing time is also taken into account. See Comparison with other sys-

tems subsection.

Q: A particular attention to articles (a and the) should be given, particularly

when they are missing (for instance l.339: it uploads the image to the shared

memory; l.343: the video streaming service checks the shared memory cotents).

A: We tried to take it into account.

Q: Bullets should be avoided in an article (it is not a presentation); a table

is often a good replacement (l 405 to 437 and l.420 to 427)

A: Fixed

3.14

Some remarks on Figures:

Q: All the figures come at the end of the article and are not included in

proper way in the article, which makes things difficult to follow.

A: This is requirement of the journal to place all figures separately from the

main text.

Q:Figure 1: I would let the mesh behind in B (as it is done in C). Moreover,

the frame used for the quadtree should be marked as such in A, just to be sure

that all Figures from A to F are at the same dimension of representation (which

does not seem to be the case).

A: In all subfigures the scale is the same. The mesh behind the quadtree

has been added. Notice that the bounding box of the quadtree is greater than

that of the original mesh by some “padding” and is obtained by including points

r + ns, where r is the position of a node of refined boundary mesh, n is the

unit vector of outer normal vector to the boundary face, and s is the size of the

refined boundary face. Since in subfigure E we use 10x boundary refinement,

and in subfigure B we use no (1x) boundary refinement, the quadtree in B has

10x larger padding than in E. It is correct and I am not sure that we need to

clarify it in the text. The need for the padding yields from the need to have a

field whose zero-valued isosurface approximates domain boundary.

Q: Figure 2: I think the mesh chosen for illustrating a bit hard to follow

and would let it by transparency under the different quadtrees generated.

A: Done

Q: Figure 3: I would add colors to help following the octree, or / and draw

the graph tree corresponding.

A: The Figure was removed due to restyling of Sparse 3D mipmapping

section.

Q: Figure 4: having the mesh by transparency under the quadtrees generated

would help.

A: The Figure was removed due to restyling of Sparse 3D mipmapping

section.

Q: Figure 5: It differs only from one node from Springer. A legend indicating

nodes signification directly on the Figure it self would help (and not in the

Legend below the Figure). The Legend below figure 5: the second B should be

C.

A: The Figure was removed due to restyling of Sparse 3D mipmapping

section.

Q: Figure 8: two inversions: Numerical solution (oi inverted) and Web

server: visualization controller (rt inverted)

A: Fixed

Q: Figure 9: ReVisE web interface and not ReVesE in the legend

A: Fixed

Q: Suggestions of figures: A figure similar to Figure 3 of [12] would clarify

lines 126-127.

A: Now we reduced the section Sparse 3d mipmapping and it is more depen-

dent on content of [12]. Therefore in order to not overload the article’s content

it is intended that the reader will refer to [12] for more information.

Q: A figure might help to clarify l.154 - 162.

A: Added Fig 3. An example of quadtree and corresponding metadata.

3.15

Mispelling / other remarks:

Q:l.403: preformance instead of performance

A: Fixed.

Q: l.345: what information does the sentence bring?

A: Removed.

---

## [Decision Letter · Decision Letter 1]

17 May 2021

PONE-D-20-33961R1

ReVisE: Remote visualization environment for large numerical simulation datasets

PLOS ONE

Dear Dr. Orlov,

Thank you for submitting your manuscript to PLOS ONE. After careful consideration, we feel that it has merit but does not fully meet PLOS ONE’s publication criteria as it currently stands. Therefore, we invite you to submit a revised version of the manuscript that addresses the points raised during the review process.

The improvement of this revised version is undeniable yet both reviewers pointed remaining minor weaknesses in the discussion regarding the comparison of ReVisE with other software, as well as in explaining interactivity. Please make sure these details are addressed as indicated by reviewers.

We look forward to receiving your revised manuscript.

Kind regards,

Frederique Lisacek

Academic Editor

PLOS ONE

Journal Requirements:

Reviewers' comments:

Reviewer's Responses to Questions

**Comments to the Author**

1. If the authors have adequately addressed your comments raised in a previous round of review and you feel that this manuscript is now acceptable for publication, you may indicate that here to bypass the “Comments to the Author” section, enter your conflict of interest statement in the “Confidential to Editor” section, and submit your "Accept" recommendation.

Reviewer #2: All comments have been addressed

Reviewer #3: All comments have been addressed

2. Is the manuscript technically sound, and do the data support the conclusions?

Reviewer #2: Yes

Reviewer #3: Yes

3. Has the statistical analysis been performed appropriately and rigorously? 

Reviewer #2: Yes

Reviewer #3: N/A

4. Have the authors made all data underlying the findings in their manuscript fully available?

Reviewer #2: Yes

Reviewer #3: Yes

5. Is the manuscript presented in an intelligible fashion and written in standard English?

Reviewer #2: Yes

Reviewer #3: Yes

6. Review Comments to the Author

Reviewer #2: I really enjoyed the reading and the improvement in quality since the first reading. From the remarks made during the first review, most of them have been lifted or answered. I still have one major remark and some additional minor remarks.

Introducing the baseline comparison suggested during the first review improves the Evaluation part. Nonetheless, from your comments: "It should be noticed that this comparison relates to the playing of time history. If one does pan/zoom/rotate in ParaView without changing time frame the volume rendering is performed very fast.", I have a question: What does happen with Revise in the pan/zoom/rotate case? Do you have additional evaluation of this point compared to paraview, such that the reader can understand all the advantages and potential drawbacks, are generally there is not only one side of the coin.

More anecdotically, preparing the dataset to be in an acceptable format for the system it is going to be read is required but I would not labelled it as a pre-processing stage as you need to do for ReVise.

Additionally, I would have expected a figure similar to Figure 11 to compare the two renderings (the one achieved with paraview and the one achieved with Revise for the hump model.

Additionally, I would put this evaluation with the corresponding dataset on a downlable virtual machine for both Paraview and Revise system in order they can be run out of the box for reviewers and readers.

Some minor additional remarks:

l. 15: you start by speaking of NVIDIA Index and Sight. NVIDIA Index is then presented in details in paragraph in l. 28 ... but no more mention of Sight.

l.59-61: I would reformulate the answer to the question in a less assertive mode, because nothing is proven at this stage, by putting:

In this paper, we are going to show that it is in fact not needed, otherwise the answer asks for explanation, which is mainly what you are going to present in the article.

I know you have mentioned that it is an editor constrain but Numbered sections are definitely needed to refer to Sections and Subsections (the Editor mentioned I should put it in my review comments). Additionally, to refer to a Section or Subsection, the rules is similar than the one applied for figures, i.e. the section name should be prefixed by Section / Subsection.

Avoid of ... of... in a sentence: it is often avoidable by reverting (3 in one sentence line 95 - 96 for instance, and also l.401, but certainly on other places)

Presenting an additional idea goes with three em-dashes: https://www.uhv.edu/university-college/student-success-center/resources/a-d/dashes-use/#:~:text=There%20are%20actually%20three%20different,writing%20to%20an%20old%20friend.

This would be appropriate in l.111-112, l.116-117, l.312, l.418-419, l.497 for instance.

Websites reference should never been put in a bibliography section but as bottom page notes, unless it is a requirement of the Editor (ref 30, 31, 1, 5, 6) as they are not proper bibliography references.

Some bullet (or telegraphic) style paragraphs remain: it is never a good idea to put more than one semi-column in a sentence. Additionally, each sub-sentence should have a subject and verb in this -case. (l. 17-20, l.86-91, l.73-77, l.296-299)

l. 158: roams instead of walks is maybe more appropriate

l. 162 - l.164: on the one hand is needed in place of importantly. End of line 161: I would add: The element size s is bounded between two values s_min and s_max. On the one hand, the smallest value s_min ...

l. 177: what is a "typical" visualization? As typical, depends mostly on the level of details you want to achieve.

l. 233 ... maybe: The interested reader can refer to [] on the way of addressing such elements and for further details.

[13] should be accessible on Arxiv, as it is intensively refered.

l. 685: to distribute

Articles should be checked, particularly when nouns are singular (here an article is often needed) as their non usage---named Zero-marking---is mostly dialectal english or reserved to particular cases.

Reviewer #3: This is a good paper. It is written well and clearly. The system that the authors developed is well described and references to the code and data are provided. It is indeed a nice system that combines various modern technologies and an octree implementation for volume rendering. The authors were objective in their comparison to ParaView.

Where the paper is weaker is in the comparison section. The authors correctly identify the performance bottleneck in ParaView as the IO time. They should not have stopped there and should have explained more thoroughly how ReVisE overcomes this issue.

Furthermore, it would have been good to better compare interaction performance rather than mainly focus on animation. Similarly for IndeX and OSPRay comparisons. Although this is not necessary to accept this paper IMO.

7. PLOS authors have the option to publish the peer review history of their article (what does this mean?). If published, this will include your full peer review and any attached files.

Reviewer #2: No

Reviewer #3: No

---

## [Author Response · Author response to Decision Letter 1]

1 Jul 2021

Please find answers to reviewers' comments in the file "response_to_reviewers_2.pdf".

---

## [Editor Report · Decision Letter 2]

5 Jul 2021

PONE-D-20-33961R2

ReVisE: Remote visualization environment for large numerical simulation datasets

PLOS ONE

Dear Dr. Orlov,

Thank you for submitting your revised manuscript to PLOS ONE. The only motivation for requesting a last minor revision is to answer your own questions regarding reviewer2's comments. To speed up the process I took the liberty of contacting directly this reviewer to get the needed explanations and here are his answers:

1) The reference to "three em-dashes" applies to a Latex option. The reviewer is suggesting that each time you have side comments in brackets as for instance in this sentence (line 123-124 revised version):

"Suppose there is an original mesh (unstructured or structured, consisting of tetrahedra or hexahedra — does not matter), and a scalar field specified at mesh nodes."

you should use the "three em-dashes" option of Latex ( \\textemdash) and write:

"Suppose there is an original mesh---unstructured or structured, consisting of tetrahedra or hexahedra, it does not matter---, and a scalar field specified at mesh nodes."

It is ultimately your choice. This is only a suggestion.

2) Regarding your questions about publishing rights regarding your Arxiv reference. The answer of the reviewer is as follows:

"You should check the date of the end of the embargo period (generally 12 months), as you could have it on an institutional website; anyway normally you are authorized on a personal website (it should be on the agreement). You can refer to: https://www.springer.com/gp/open-access/publication-policies/self-archiving-policy"

Otherwise, your last revision matches all other requirements for publication so, please resubmit as soon as you can.

A rebuttal letter that responds to these last two items after the explanations of reviewer2. You should upload this letter as a separate file labeled 'Response to Reviewers'.A marked-up copy of your manuscript that highlights changes made to the original version. You should upload this as a separate file labeled 'Revised Manuscript with Track Changes'.An unmarked version of your revised paper without tracked changes. You should upload this as a separate file labeled 'Manuscript'.

We look forward to receiving your revised manuscript.

Kind regards,

Frederique Lisacek

Academic Editor

PLOS ONE
---

## [Author Response · Author response to Decision Letter 2]

7 Jul 2021

Please find answers to reviewers in file response_to_reviewers_3.pdf

---

## [Editor Report · Decision Letter 3]

9 Jul 2021

ReVisE: Remote visualization environment for large numerical simulation datasets

PONE-D-20-33961R3

Dear Dr. Orlov,

We’re pleased to inform you that your manuscript has been judged scientifically suitable for publication and will be formally accepted for publication once it meets all outstanding technical requirements.

Kind regards,

Frederique Lisacek

Academic Editor

PLOS ONE
---

## [Editor Report · Acceptance letter]

16 Jul 2021

PONE-D-20-33961R3 

ReVisE: Remote visualization environment for large numerical simulation datasets 

Dear Dr. Orlov:

I'm pleased to inform you that your manuscript has been deemed suitable for publication in PLOS ONE. Congratulations! Your manuscript is now with our production department. 

Kind regards, 

on behalf of

Dr. Frederique Lisacek 

Academic Editor

PLOS ONE